# Model-based assessment of climate change impact on inland flood risk at the German North Sea coast caused by compounding storm tide and precipitation events

Helge Bormann[1], Jenny Kebschull[1], Lidia Gaslikova[2], Ralf Weisse[2]

[1]Jade University of Applied Sciences, Ofener Strasse 16/19, 26121 Oldenburg, Germany
[2]Helmholtz-Zentrum Hereon, Max-Planck-Straße 1, 21502 Geesthacht, Germany

*Correspondence to*: Helge Bormann (helge.bormann@jade-hs.de)

**Abstract.** In addition to storm tides, inland flooding due to intense rainfall becomes an increasing threat at coastal lowlands. In particular, the coincidence of both types of events poses great challenges to regional water boards since their technical drainage capacities are limited. In this study, we analysed historical data and scenario-based simulations for the gauge Knock near Emden at the German North Sea coast. The evaluation of observed inland flood events shows that mainly moderate storm tide series in combination with large-scale, intense precipitation led to an overload of inland drainage systems, whereas the highest individual storm tides or precipitation events alone could be handled well. A pro-active risk management requires climate projections for the future. Therefore, a hydrological and a hydrodynamic ocean model were set up and driven by the same climate simulations to estimate future drainage system overloads. The evaluation of the simulations for the control period of two climate models confirms that the models can reproduce the generation mechanism of the compound events. The coincidence of storm tides and precipitation leads to the highest drainage system overloads, while system overload is also caused by intense rainfall events alone rather than by storm tides without intense precipitation. Scenario projections based on two climate models and two emission scenarios suggest that the intensity of compound events of rainfall and storm tides will increase consistently against the background of mean sea level rise for all investigated climate projections, while simulated system overload is higher for RCP8.5 compared to RCP2.6 scenario. Comparable to the past, future compound events will cause more potential damage compared to single extreme events. The model results indicate an increasing frequency and intensity of inland drainage system overloads along the North Sea coast if timely adaptation measures will not be taken.

## 1 Introduction

Drainage management at flat coasts is a traditional challenge to enable the use of the dike hinterland for human activities such as agriculture, settlements or trade (Titus et al., 1987; Ritzema and Stuyt, 2015; Waddington et al., 2022). The inhabitants of the Northwest-German marsh area along the North Sea started building dikes and sea walls more than 1000 years ago to protect the productive landscape against storm tides (Behre, 2002). Due to the humid climate and the corresponding rainfall excess of inland areas, concurrent development of an efficient drainage system was necessary (Bormann et al., 2020). Tidal gates and

pumping stations were built to convey the excess water from the hinterland to the sea. Such drainage management has been improved over the past decades (Spiekermann et al., 2018, 2023), keeping all areas productive during the year and protecting low-lying coastal areas against inland flooding. Similar systems for coastal protection and inland drainage were also developed in other low-lying coastal areas around the world, such as the Netherlands (van Alphen et al., 2022; Ritzema and Stuyt, 2015), the United States of America (Titus et al., 1987), Australia (Waddington et al., 2022), or New Zealand (Kool et al., 2020).

Climate change projections suggest that global mean sea level will continue to rise until the end of the 21st century and beyond (IPCC, 2021). Coastal areas therefore are identified as highly threatened areas (IPCC, 2022). Low-lying coasts are particularly vulnerable because water hazards may come from two sides (van den Hurk et al., 2015; Kool et al., 2020; Xu et al., 2023). Both, sea level rise and intense rainfall events will have an increasing impact on flooding until the end of the century. Recent studies indicate that climate change also might intensify the pronounced seasonality in runoff generation along the North Sea

coast (Bormann and Kebschull, 2023; Bronstert et al., 2023), leading to increasing drainage demands in the future if drainage standards are kept at least at the status quo.

    However, water boards along the North Sea coast are increasingly operating their systems at the edge of their capacity as a consequence of increasing climate variability and extreme events (Spiekermann et al., 2018, 2023). Water boards therefore are aware that the effects of climate change will place even greater demands on drainage infrastructure in the future (Ahlhorn et

al., 2018). Coastal water boards are responsible for the regulation of the water flow in the regional drainage networks, consisting of canals, ditches, tidal gates, and pumping stations. They guarantee both water drainage in wet periods, but also regulation of discharge in dry periods. In addition to the individually occurring events such as storm tides or intense precipitation, compound events in particular pose a special challenge for flood protection and inland drainage. If, for example, storm tide and intense rainfall events occur at the same time, even a combination of moderate single events often leads to an

overload of the inland drainage system (Santos et al., 2021; Kool et al., 2020; van den Hurk et al., 2015). According to Pugh and Woodworth (2014), a storm tide is defined as the sum of a storm surge and the astronomical tide. At the German North Sea coast of Lower Saxony, such an event was last observed in February 2022, when a series of storm depressions (Ylenia, Zeynep, Antonia) was accompanied by several days of intense rainfall. Consequently, regional flooding occurred due to drainage system overload.

Climate change adaptation and flood risk management must explicitly consider the flood generation mechanism of such events (van den Hurk et al., 2015), especially if climate change will intensify as projected by the IPCC (2021, 2022). Most of the available studies focus on the coincidence of storm tides and high regional river discharges. For example, Heinrich et al. (2023) investigated the coincidence of storm tides and high river discharge at European coasts. While they found a significantly increased likelihood of simultaneous storm tides and high river discharge for westward-facing estuaries, they did not identify

a frequency higher than expected by chance of such compound events for other estuaries including the northward-facing Ems estuary. Similarly, Svensson and Jones (2002) analysed the dependence of storm tides, river flow, and precipitation in the UK. They identified compound events of high river flow and storm for specific storm track directions. The dependence between high river flow and storm surge was found to be stronger during winter than in summer.

While most available studies focus on data analyses, Paprotny et al. (2020) demonstrated that large-scale hydrodynamic models are capable of representing observed compound flood events in northwestern Europe. Thus, model-based tools can be useful for the projection of climate change impacts. Bevacqua et al. (2019, 2020) projected a strong increase in the occurrence rate of compound flooding events for the future, especially in northern Europe, mainly due to the stronger precipitation as the result of a warmer atmosphere carrying more moisture. A similar result was obtained by Heinrich et al. (2023b) but was attributed mostly to future rising mean sea level. Xu et al. (2023) investigated the impact of future rainfall changes and sea level rise on compound flood risk for a coastal city in China by hydrodynamic modelling. They showed that both drivers increase flood extent, flood depth, and flood duration while future rainfall changes will have a greater impact on flood risk. However, the combined effect of future rainfall changes and sea level rise on flood risk is assumed to be larger than the sum of their individual effects.

Other climate change impact analyses focus either on the impacts of storm tides (e.g., van Alphen et al., 2022; Waddingten et al., 2022), on changes in runoff generation (e.g., Bormann et al., 2018; Bormann and Kebschull, 2023), or are based on statistical approaches (e.g., Santos et al., 2021). This latter is a consequence of the fact that the available resolution of climate projections is often insufficient to drive regional hydrodynamic ocean models to represent water level dynamics including storm tides. After all, they require regional wind fields in high temporal resolution. Similarly, process-based runoff generation models require precipitation intensities in sufficient resolution.

So far, no scenario-based climate change impact assessment is available projecting the above-described compound events causing local to regional flooding at shallow coasts by coupling ocean and hydrological models. While on the global scale such impact is neglected, local to regional studies focus either on data analyses (e.g., van den Hurk et al., 2015) or combine explicit modelling of the terrestrial hydrology with assumptions on sea level rise and storm surges (e.g., Xu et al., 2023).

In this study, we fill this gap by driving a runoff generation model and a regional hydrodynamic ocean model with the same regionalized climate scenarios in hourly temporal resolution, enabling consistent coupled projections of compound events. We use climate projections of two climate models quantifying the effects of two different emission scenarios and analyse this small ensemble with regard to change signals in the impact of storm tides and runoff generation on the overload of a regional drainage system at the German North Sea coast.

## 2 Material and methods

### 2.1 Target area

The target area of this study is located in the north-western part of East Frisia (northern Germany) and includes the area of the water board Emden (EEVE, Figure 1). The area of the water board Emden is bordered by the Ems estuary, the Dollart, and the North Sea. The water board has an area of 465 km², from which 1/3 is located below sea level (Spiekermann et al., 2018). The landscape is dominated by marsh soils, mainly used for dairy farming (grassland). But the region also has residential and commercial areas, and tourism plays an important role in the regional economy.

Since permanent and reliable drainage is the prerequisite for the settlement and use of the area, the water board maintains a watercourse network of 1,100 km in length, which conveys the runoff to the two tidal gates and pumping stations Knock and Greetsiel. Depending on the sea water level, either free drainage is possible (currently approx. 1/3 of the drainage volume) or pumping is required (currently approx. 2/3 of the drainage volume). In the case of high sea water levels, the pumping capacity drops significantly due to the increasing geodetic head (Bormann et al., 2023; for details see section 2.4.3).

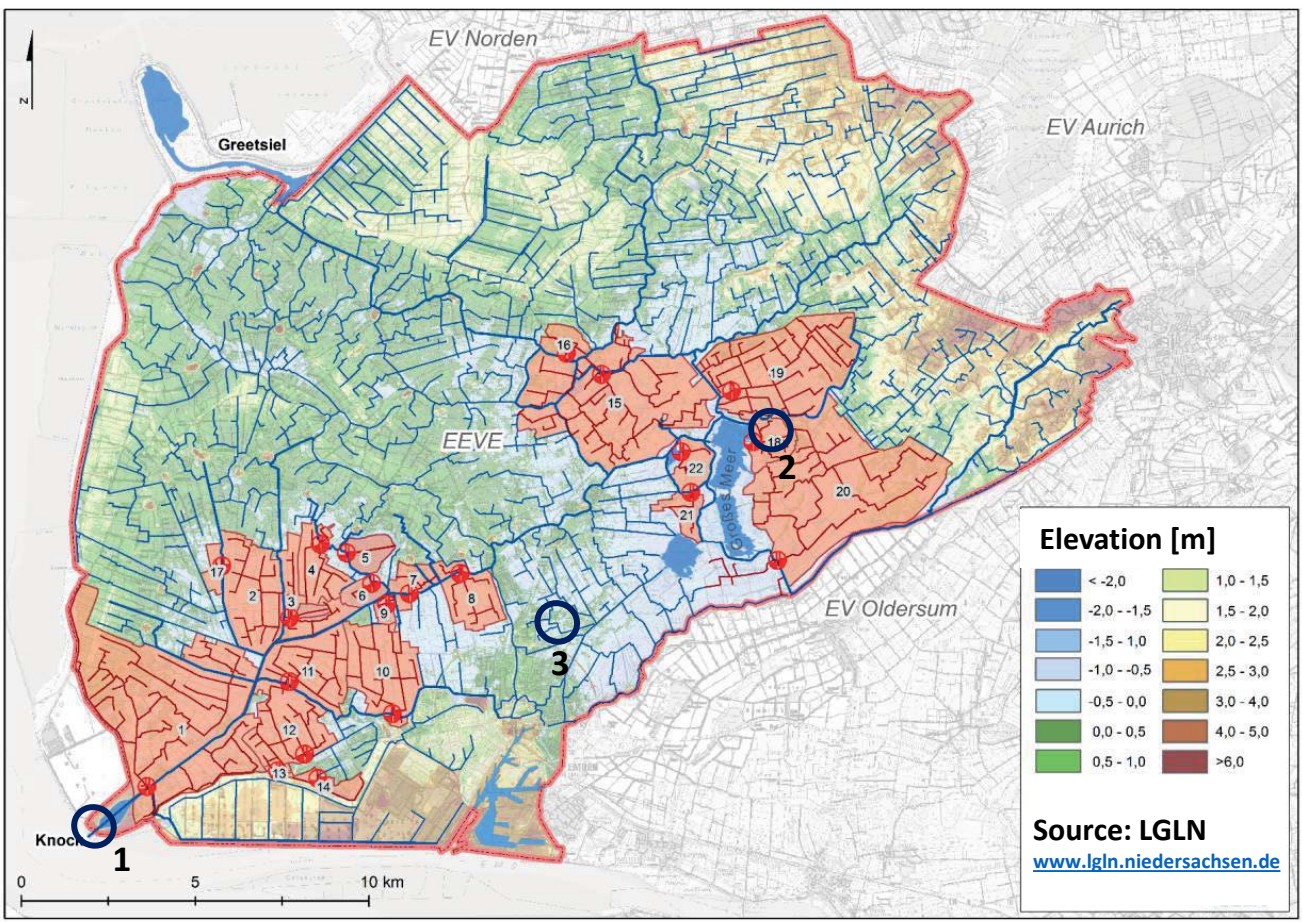

**Figure 1: Topography of the water board Emden (EEVE; area within the red border; elevation in [m] above NHN=standard elevation zero). Orange areas are 22 low-lying pump-areas (black numbers), blue linear structures represent the drainage system. Black circles locate the measurement stations: 1: gauge Knock; 2: gauge Bedekaspel; 3: DWD weather station Emden. Data source: extract from the basic geodata of the Lower Saxony surveying and cadastral administration (LGLN; www.lgln.niedersachsen.de).**

Specific inland flood risk is caused in periods when storm tides and intense rainfall coincide. In such a situation high runoff generation and reduced pumping capacities can lead to drainage system overload and inland flooding. The effect of high River

Ems discharge is negligible compared to the impact of storm tides due to the wide Ems estuary at gauge Knock and due to the time lag between high precipitation and river runoff caused by the catchment size (>13,000 km²). However, water boards located upstream are impacted by high Ems river discharge, as well.

## 2.2 Data

For the analyses of historic events, a regional 20-year time series (2000-2019) of sea water level, inland water level, and weather data was available for the water board Emden in high temporal resolution:

- Sea water levels in 1-minute resolution at gauge Knock (location 1 in Figure 1; source: state agency NLWKN; www.nlwkn.niedersachsen.de);
- Inland water levels in 15 minutes resolution at gauge Bedekaspel (location 2 in Figure 1; source: Emden water board; www.entwaesserungsverband-emden.de);
- Precipitation in daily resolution at station Emden (location 3 in Figure 1; source: German Weather Service; www.dwd.de).

As forcing for the impact modelling, four data sets from the Euro-CORDEX initiative were available with the sufficient temporal resolution. The data sets provide regional climate projections for Europe at 12.5 km (0.11°) resolution (Jacob et al. 2014). For these data sets, the regional climate model REMO (Jacob et al. 2007) in two different versions (REMO2009 and REMO2015) was driven by the output of two global climate models (HadGEM2-ES; Jones et al. (2011), and MPI-ESM-LR; Giorgetta et al. (2013)) being part of the CMIP5 ensemble (Taylor et al., 2012). For both regionalizations, one realization with hourly model output was available for a historic period, and two emission scenarios (RCP8.5 and RCP2.6) in the following combinations:

- HadGEM2-ES, regionalized by the regional climate model Remo2009 (1950-2099; RCP2.6 and RCP8.5) referred to as HadGEM in the following, and
- MPI-ESM-LR, regionalized by the regional climate model Remo2015 (1950-2100; RCP2.6 and RCP8.5) referred to as MPI in the following.

Bias correction was carried out for the simulated temperature and precipitation of both dynamic climate models. A monthly linear scaling (Shrestha et al., 2017) was applied to the control period (1971–2000) to correct a long-term overestimation of precipitation and temperature. To reproduce seasonality in precipitation and temperature correctly, monthly specific bias correction factors were applied to the model data for all months of the year (for details see Ley et al., 2023; Bormann and Kebschull 2023).

## 2.3 Classification and determination of compound events

Compound events are defined as a combination of processes (climate drivers and hazards) leading to a significant impact on one system (Zscheischler et al., 2018). The IPCC (2012) describes compound events as combinations of (1) two or more extreme events occurring simultaneously or successively, (2) combinations of extreme events with underlying conditions that

amplify the impact of the events, or (3) combinations of events that are not themselves extremes but lead to an extreme event or impact when combined. The contributing events can be of similar (clustered multiple events) or different type(s) (Seneviratne et al., 2012).

Based on these definitions Zscheischler et al. (2020) suggested a typology of meteorological compound events. They distinguish between

1.      Multivariate compound events: simultaneous occurrence of *different events* affecting the same system,

2.      Spatial Compound Events: occurrence of events in *different regions* affecting the same system,

3.      Temporal compound events: several *consecutive events* affecting the same system, and

4.      Preconditioned compound events: events that occur only under *certain conditions*.

In this study, the multivariate compound events resulting from the coincidence of storm tide series and intense precipitation on the North Sea coast are investigated for the Emden water board (East Frisia). Practical experience of the water board shows that system overload at the North Sea occurs, especially when intense precipitation falls over a longer time during a period in which the pumping capacity is reduced by high sea levels (Spiekermann et al., 2018, 2023). In the course of such events the drainage system overload is exacerbated by the technical limitation of the pumping capacity installed in the dike line.

Compound events in the observations were identified following the approach suggested by van den Hurk et al. (2015) by selecting the 15 largest events according to sea level (mean sea level over a 5-tide period), precipitation (antecedent 3-day precipitation sum) and highest inland water levels (maximum daily value). This corresponds to a characteristic three-day period as storm tide series of three days duration were already observed in the region in the past years (e.g., in February 2022). For adaptation-planning, regional stakeholders therefore demand an estimation of the impact of compound events with 3-day duration (Spiekermann et al., 2023). The water boards aim at a constant water level of the inland water system to guarantee a permanent use of the area. Thus, a positive deviation from the target water level can be interpreted as a system overload. While van den Hurk et al. (2015) compared the 20 largest inland water levels to the highest 10 wind-induced surges and precipitation events, we extracted 15 events each as a compromise between limited data availability (20-year time series of observations) and a minimum number of events required for evaluation.

For the model-based analyses, compound events were identified by an adjusted method, since inland water levels could not be simulated directly. This is due to the complex drainage system and the anthropogenic regulation of the hydraulic head, while regulation is not documented and is not always rule-based. Instead of water levels, system overloads [in million m³] were calculated as the sum of the current system overload at a distinct day and runoff generation, minus the pump-capacity of that day (which depends on average sea level; for details see the following section on the model-set-up). By selection of a characteristic three-day period, a potential delay between precipitation and system overload ("concentration-time") is taken into account. Similar to the analysis of the observed data, the 15 largest events were selected for the highest sea levels (average sea level over a period of five tides) and precipitation events (antecedent 3-day precipitation sum). 15 events with the highest magnitudes of the simulated system overloads of the inland drainage system were used as a proxy for high inland water level events.

To describe the compound events with regard to their drivers, the maximum of the 15 selected largest sea levels and precipitation events each are called *extreme* events, while the remaining 14 events are called ***high*** storm tides / ***intense*** precipitation. Further events which are lower than the 15 highest selected are called ***moderate*** events. While such definition may lead to the fact that, under climate change conditions, an ***extreme*** event from the past could be only classified as an ***intense*** event in a future period, it enables to assess the contribution of the individual drivers to a compound event with regard to the ensemble of events in the respective period.

## 2.4 Model set-up

While the analysis of compound events based on observations could be carried out straightforward according to the literature (van den Hurk et al., 2015), for the evaluation of climate change projections a new model set-up was necessary to represent the generation of multivariate compound events. Extreme and intense precipitation events and high water levels of storm tides were derived from simulations directly, but due to the complexity of the coastal drainage system, inland water levels could not be simulated for the water board area; instead, system overload was estimated as follows (Figure 2). A storage-approach represents the volume of the coastal drainage system. Runoff-generation simulated by the SIMULAT water-balance-model (Diekkrüger and Arning, 1995; Bormann, 2008) is the only input, and pump-capacity the only output term. We assume that there is no exchange of water between the area of the Emden water board and the neighbouring water board areas and that there is no limitation of lateral water flow within the water board area through canals and ditches. The time-variable drainage capacity then depends on the sea level simulated by the regional hydrodynamic ocean model TRIM-NP (Kapitza et al. 2008), the inland water level (assumed to be constant at the current target level), and the parameters of the pumps. Since both models (SIMULAT and TRIM-NP) are driven by the same climate projections, consistent representations of possible future conditions are simulated.

An application of this methodology to the observed data was not carried out due to the limited temporal resolution of precipitation data. Since precipitation is available only in daily resolution, a disaggregation would have been required introducing additional uncertainty.

### 2.4.1 SIMULAT model: Runoff-generation

The physically based hydrological model SIMULAT (Diekkrüger and Arning, 1995; Bormann, 2008) is a continuous hydrological SVAT (soil-vegetation-atmosphere-transfer) scheme, initially developed to simulate local-scale hydrological processes. Basic equations included in SIMULAT are the Richards' equation to calculate soil water flux and the Penman-Monteith equation for potential evapotranspiration (PET). Actual evapotranspiration (ETA) is calculated from PET taking into account the actual soil moisture status (approaches of Feddes et al. (1978) for transpiration and Ritchie (1972) for evaporation). Further processes considered by SIMULAT are the separation of rainfall into surface runoff and infiltration (performed by a semi-analytical solution of the Richards' equation (Smith and Parlange, 1978)), interflow (based on Darcy's law), groundwater recharge, and snowmelt (degree-day approach). A plant growth model is not included; instead, the average seasonal

development of plant parameters necessary for the Penman-Monteith equation is estimated by linear interpolation of values given from the literature (see Bormann and Kebschull, 2023). Soil parameters are derived by the pedotransfer-function of Rawls and Brakensiek (1985). At the scale of hydrological response units (HRU) derived for the Emden water board by Bormann et al. (2018) based on the available spatial data sets, runoff generation is calculated by accumulating all three runoff components (surface runoff, interflow, and groundwater recharge) for each daily time step. At the water board scale, runoff generation is calculated for all HRU and afterwards weighted per unit area. Runoff routing is not considered since hydraulic gradients are mainly affected by the operation of the drainage system and are therefore temporarily variable. We assume that the concentration time is considerably smaller than the maximum duration of the events analyzed in this study (3 days). SIMULAT was successfully calibrated and validated for the Emden water board (Bormann et al., 2018), and for neighboring water boards (Bormann and Kebschull, 2023) by comparing runoff generation simulated by the model against the estimated drainage rates.

For the climate change impact analysis, SIMULAT is driven by hourly climate variables of the regionalized climate models MPI and HadGEM, namely air temperature, air humidity, wind speed, global radiation, and precipitation. As output, SIMULAT calculates daily runoff generation rates.

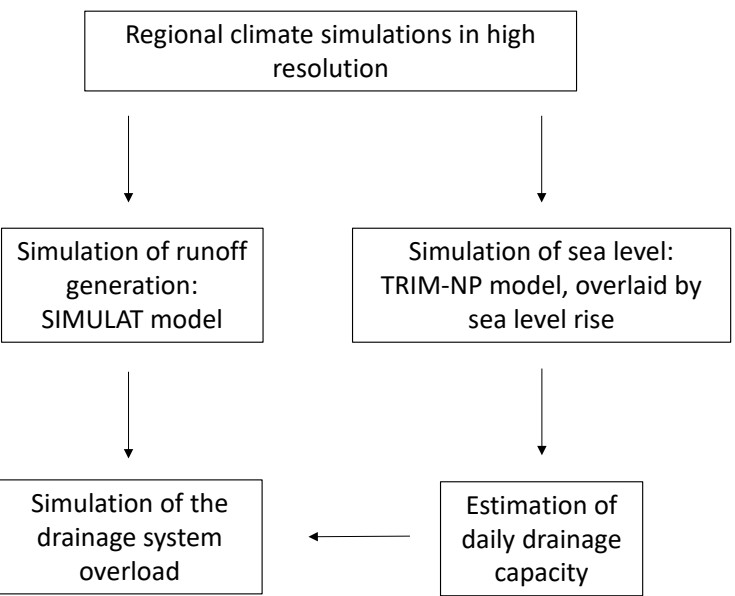

**Figure 2: Model chain for the projection of future compound events: Regionalized climate projections drive a regional runoff-generation model (SIMULAT) and a regional hydrodynamic ocean model (TRIM-NP); estimation of daily drainage capacity from sea water levels enables the comparison against runoff generation, resulting in projections of drainage system overload.**

### 2.4.2 TRIM-NP model: Total sea level

For the high-resolution modelling of water levels in the German Bight and the attached estuary of the river Ems, the
hydrodynamic numerical model TRIM-NP (Casulli et al 1998, Kapitza 2008) is used. TRIM-NP is a 3D finite-difference
model, which solves the Reynolds-averaged Navier-Stokes equations on a Cartesian grid. It allows for wetting and drying. For
the simulations in this study, the model is used in a 2D barotropic mode with nested grids. The coarsest grid with 12.8 km
resolution covers the northeastern Atlantic, North Sea, and Baltic Sea. Three further grid refinements are nested one-way
towards the 1.6 km resolution over the German Bight. The FES tidal signal (Layard et al 2006) is applied at the lateral
boundaries of the coarsest grid. For the climate change impact analysis, TRIM-NP is driven by hourly 10m-height wind
components and sea level pressure from the regionalized climate models MPI and HadGEM. The output sea level variations
for the period 1950-2100(2099) are stored with 20 min resolution for the German Bight (Gaslikova, 2023). For further model
details and applicability for the climate change scenarios see also Gaslikova et al (2013). The resulting high-frequency sea
level variations are additionally superimposed by regionalized long-term projections of the mean sea level rise (IPCC, 2021).
In particular, for the period 2020-2100 the median of the regional SSP1-2.6 and SSP5-8.5 projections for Delfzijl from the
IPCC AR6 Sea-Level Projection Tool (Garner et al., 2021; Fox-Kemper et al., 2021) are used together with the climate
projections RCP2.6 and RCP8.5 respectively. For the sake of clarity all combinations of modelled storm tides and mean sea
level will be referred to as either RCP2.6 or RCP8.5 further on. The dataset is complemented with the observed annual mean
sea level at Norderney (WSV, 2021) for the historical period 1971-2019.

### 2.4.3 Drainage capacity

The drainage capacity of a coastal outlet structure consisting of a tidal gate and a pumping station (such as station Knock for
the Emden water board; Figure 1) depends on both individual drainage capacities which apply as a function of total dynamic
head between inland water level and sea level. While for low sea level gravity-driven flow through the tidal gate is possible,
water needs to be pumped against high sea level.

Pump capacity depends on total dynamic head which is defined as the work to be done by a pump, per unit weight, per unit
water volume. Based on an estimation of gravity-driven flow through the tidal gate (provided by the state agency; NLWKN)
and the pump parameters (provided by the Emden water board), a drainage capacity function was derived for the station Knock.
Estimation of the pump capacity is based on the pump characteristics and is done for each blade position, which is adjusted in
intervals of 50 cm hydraulic head. Above a hydraulic head of 3.75 m, the pumps have to be switched off. A 3rd-order polynomial
function was fitted by the least squares approximation for the estimated values (see Figure 3). The drainage system is managed
in a way, that the inland water levels remain as constant as possible. The deviations from the intended inland water level occur
in case of system overload and will be analysed in more details in the next section. Although this unwanted increase of the
inland water level in the order of a few decimetres is critical for flood risk analysis, it does not significantly change the pump
capacity. As can be seen from the Figure 3, a change of total dynamic head for several decimetres does not have a strong

impact on the pump rates. Thus, it is assumed that the hydraulic head mainly depends on the sea level outside the pump station, and pump capacity considerably decreases with increasing sea water level.

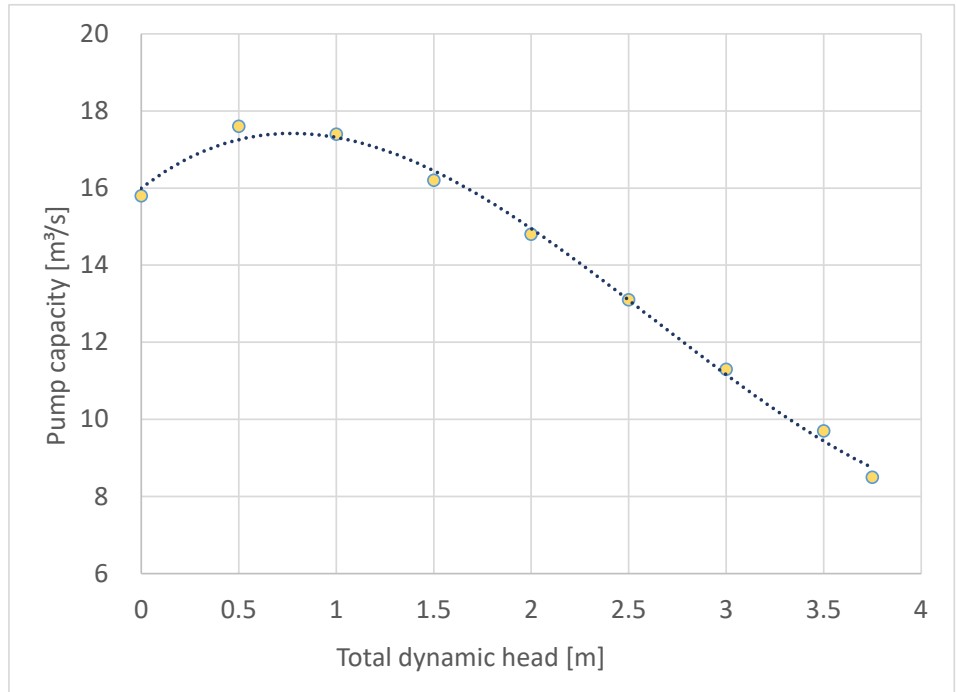

**Figure 3: Function of the pump capacity per pump at station Knock. The yellow dots are estimations of the pump capacity based on the pump characteristics, the dotted line is a corresponding 3rd order polynomial approximation.**

**3 Results**

**3.1 Identification of compound events based on historical data**

The water level in the inland water system is regulated to keep the inland water level constant. The difference between the observed and the intended (=regulated) inland water level provides a proxy for the drainage system overload. To analyze possible compound effects, the events with the 15 largest inland water anomalies were selected and plotted together with the

corresponding observed mean sea level over five tides and the antecedent three-day precipitation sum. In addition, the 15 largest sea level and precipitation events were also selected and plotted together with their observed inland water level anomalies (Figure 4). If none of the 15 highest precipitation events and 15 storm tides contributed to the 15 highest inland water levels, this would result in 45 events in total, while if all of the highest observed storm tides and rainfall events would contribute to the 15 highest inland water levels, only 15 circles would be drawn.

The analysis reveals that either the co-occurrence of high sea levels together with intense precipitation events, or the joint occurrence of moderate sea levels and intense precipitation sums lead to the largest observed deviations from the regulated

inland water level (in total nine of the 15 largest events: orange bubbles with red or/and blue contour). This is in agreement with the results and conclusions presented by van den Hurk et al. (2015). Moreover, a tendency can be inferred that precipitation is a somewhat more important driver as five of the 15 highest inland water level anomalies were associated with

intense precipitation only (orange bubbles with blue contour) while sea level heights were moderate. In addition, Figure 4 reveals that sea level and precipitation do not represent the sole drivers of inland water anomalies as some of the events with rather similar combinations of precipitation and sea levels were associated with rather different inland water heights. Thus, assuming storm tide-precipitation multivariate compound event gives insight into the overload generation mechanism. But it cannot explain the variability in the data alone. Preconditions such as antecedent soil moisture probably are an important

additional driver, but observational data on soil saturation is not available.

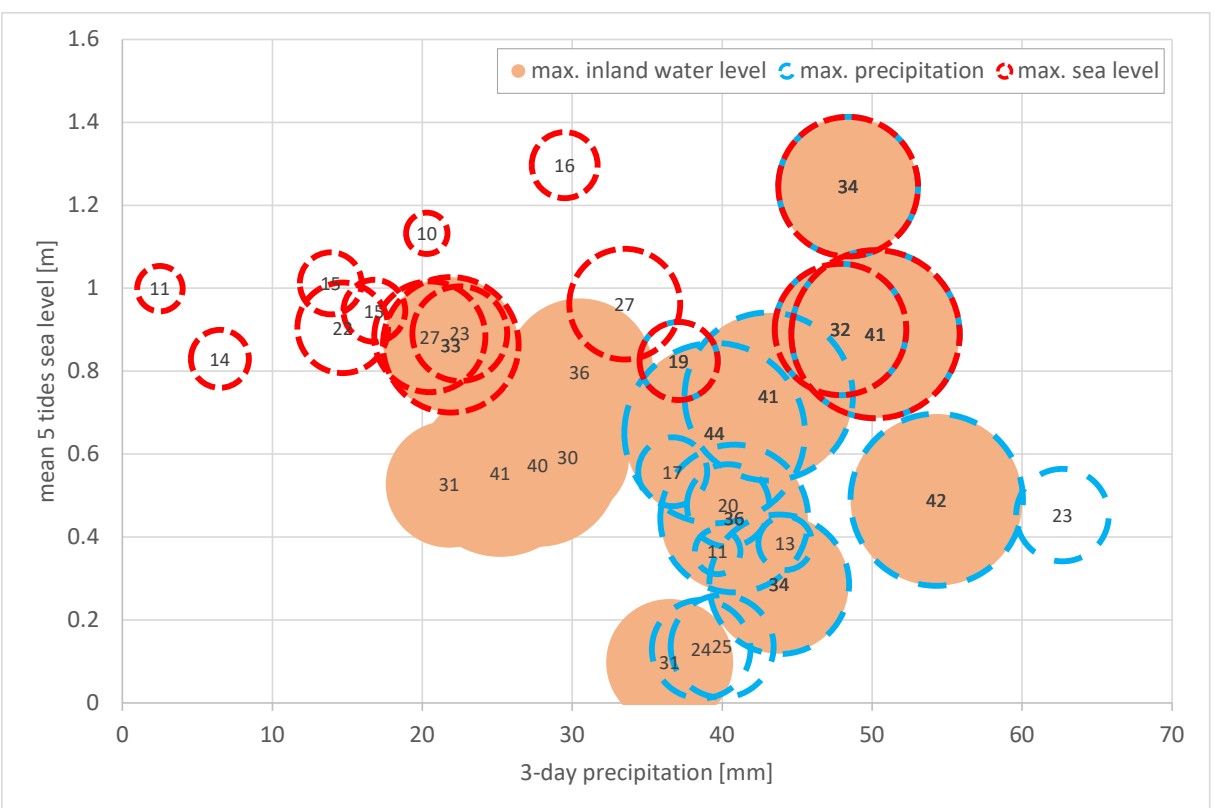

**Figure 4: Inland water level anomaly as difference from the regulated inland water level [cm] for the 15 highest sea level (mean sea level over a 5 tides period above NHN [m]; red circles) and precipitation (antecedent 3 days precipitation**

**sum [mm]; blue circles) events in the period 2000-2019. Additionally, the 15 highest inland water level anomalies in the same period are shown (orange bubbles). The diameter of the circles is proportional to the size of difference between the observed and the intended regulated inland water level (number in the circles/bubbles in [cm]).**

## 3.2 Identification of compound events based on scenario simulations

### 3.2.1 Control period of the climate simulations

The analysis of compound events from the impact models driven with data from the two downscaled climate models (HadGEM, MPI) for the control period (1971-2000) reveals that the model set-up can reproduce the structure of compounding sea level and precipitation events contributing to inland flooding as obtained from the observations. Repeating the analysis done for the observations (Figure 4) for the 1971-2000 control period yields the same three different groups of events (Figure 5), as derived from observed data (section 3.1). Note that instead of inland water level anomaly [cm], simulated drainage

system overload [million m³] was used in the analysis of the model results.

For the control period, the overlap of the 15 largest 5-tides mean sea levels to the 15 highest inland drainage system overloads (7 events for the MPI model, no event for HadGEM) is smaller compared to the highest precipitation events (8 events for the MPI model, 5 events for HadGEM). Especially for the HadGEM model, intense precipitation seems to be the more important driver contributing to high inland drainage system overloads. Largest system overloads (13.7 million m³ for MPI model, and

13.3 million m³ for HadGEM model) are neither caused by the maximum sea level nor by the largest precipitation sum. They are generated by a combination of moderate storm tides and intense precipitation. However, obviously both precipitation sums and sea levels are larger in the control simulations compared to the observational data. Such events would lead to local inundations since only about half of the calculated volume can be retained by the regional drainage system. (Spiekermann et al., 2023). The magnitude of the calculated events follows the perception of the regional stakeholders on the highest events

experienced in the past decades (Spiekermann et al., 2023), while differences in the observed water anomalies (Figure 3) may partially be due to the shorter period of observational data, compared to the simulations.

For precipitation, the overestimation by the simulations can also be caused by individual small-scale convective summer events. In reality, such events do not cover the whole water board area and therefore do not induce a system overload. However, they are overrepresented in the results of the regional climate models due to relatively coarse grid size. Therefore, the same

evaluation of compound events was repeated with a focus on events in the winter half-year, only (October to March; Figure 6).

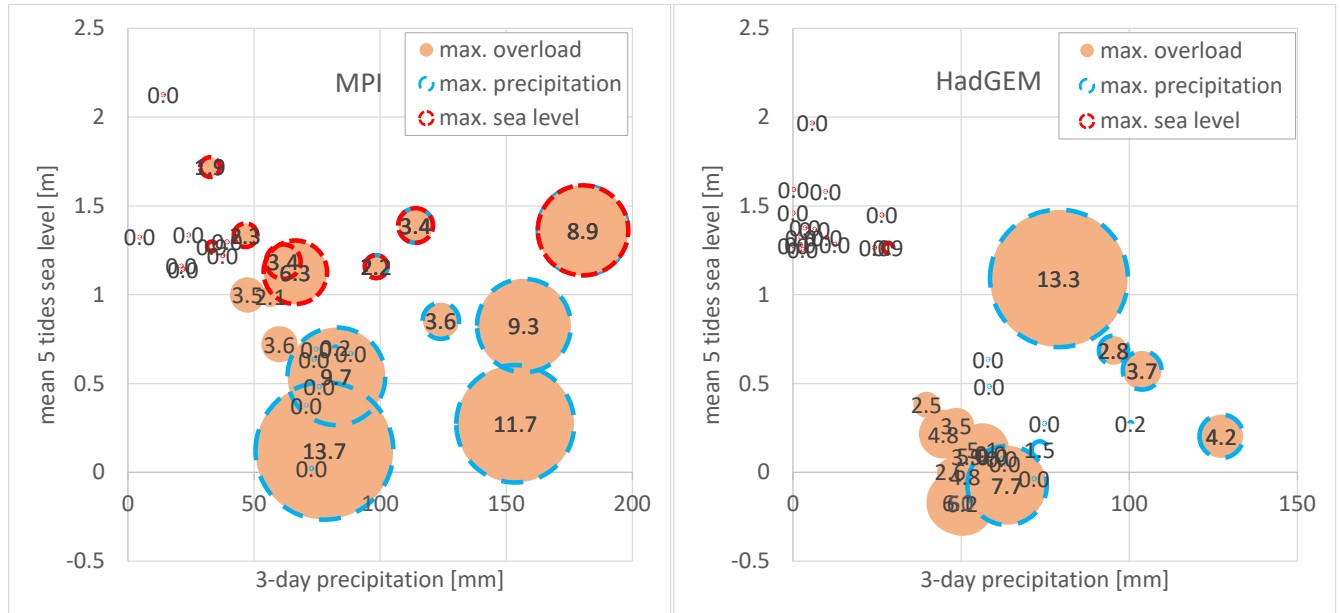

**Figure 5: Simulated drainage system overload [$10^6$ m³] for the 15 highest sea level (mean sea level over a 5 tides period above NHN [m]; red circles) and precipitation (antecedent 3 days precipitation sum [mm]; blue circles) events in the period 1970-2000 derived from the simulations driven by the MPI (left) and HadGEM (right) model data. Additionally, the 15 simulated drainage system overloads in the same period are shown (orange circles). The diameter of the circles is proportional to the magnitude of the modelled drainage system overload (number in the circles/bubbles in [million m³]).**

As expected, the winter subset resulted in considerably lower precipitation amounts, but in a similar pattern of compound events. While 4 of the highest system overloads simulated by the MPI model co-occur with extreme and intense precipitation events and high storm tides (orange bubbles with red and blue circles), the system overloads simulated by the HadGEM model are mainly caused by intense precipitation (orange bubbles with blue circles; Figure 6). None of the highest 15 storm tides out of HadGEM contributes to the highest 15 system overloads, and most of the system overloads are simulated for moderate sea levels which are more than 0.5 m lower than for MPI model. Analysing the timing of the events reveals that for MPI most of the highest 15 storm tides (11 out of 15) and precipitation events (13 out of 15) are simulated in late autumn. HadGEM also simulates most of the precipitation events in late autumn (11 out of 15), while the simulated storm tides are more equally distributed over the winter season (8 in late autumn and 7 in winter). Therefore, it is more likely for the MPI model that high storm tides and intense precipitation co-occur. With regard to the pattern of co-occurrence, MPI seems to be closer to the observations than HadGEM in terms of the number of events (see also Table 1).

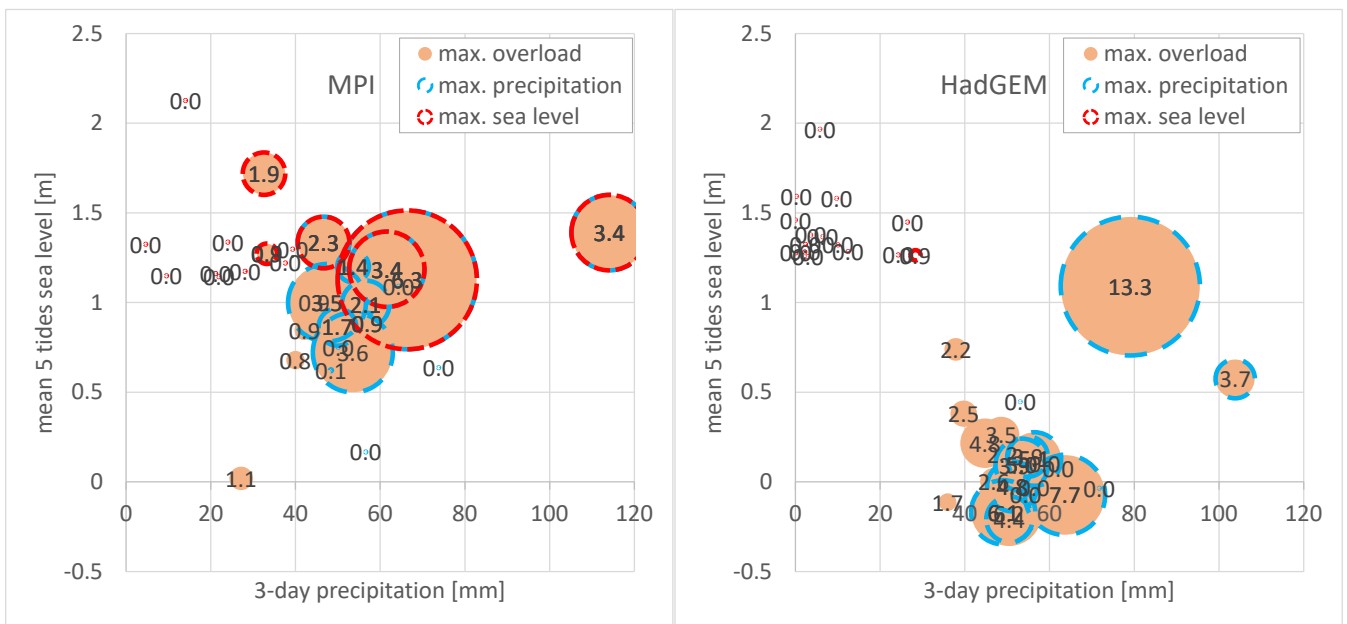

**Figure 6: As Figure 5 but for the winter half-year (October to March) only (number in the circles/bubbles in [million m³]).**

 **3.2.2 Impact of climate change on water balance terms**

All climate scenarios applied in this study feature a temperature increase until the end of the 21st century. Regional warming is projected for all seasons while the increase in temperature seems to be larger in winter than in summer. For East Frisia, the temperature rise projected by the HadGEM model exceeds that of the MPI model (Ley et al., 2023). Compared to temperature, the projected patterns in precipitation are much more variable, depending on the season, the emission scenario, the model, and the time period considered in the future. While on the annual scale, both climate models and emission scenarios considered in this study project a more humid future, change signals are model and scenario-specific on a seasonal scale (Figure 7). Here, the HadGEM model projects a stronger increase in winter precipitation for the far future (end of the 21st century) which can be expected to be a major driver of future compound events.

Also for the evapotranspiration calculated by the SIMULAT model, the projected changes depend on the climate model and the scenario. While projections based on the HadGEM model show an increase for both scenarios and periods in summer and autumn, the signal is not that clear for the climate projections based on the MPI model (figure 7). The difference in the projected evapotranspiration can be mainly attributed to the differences in the temperature projections.

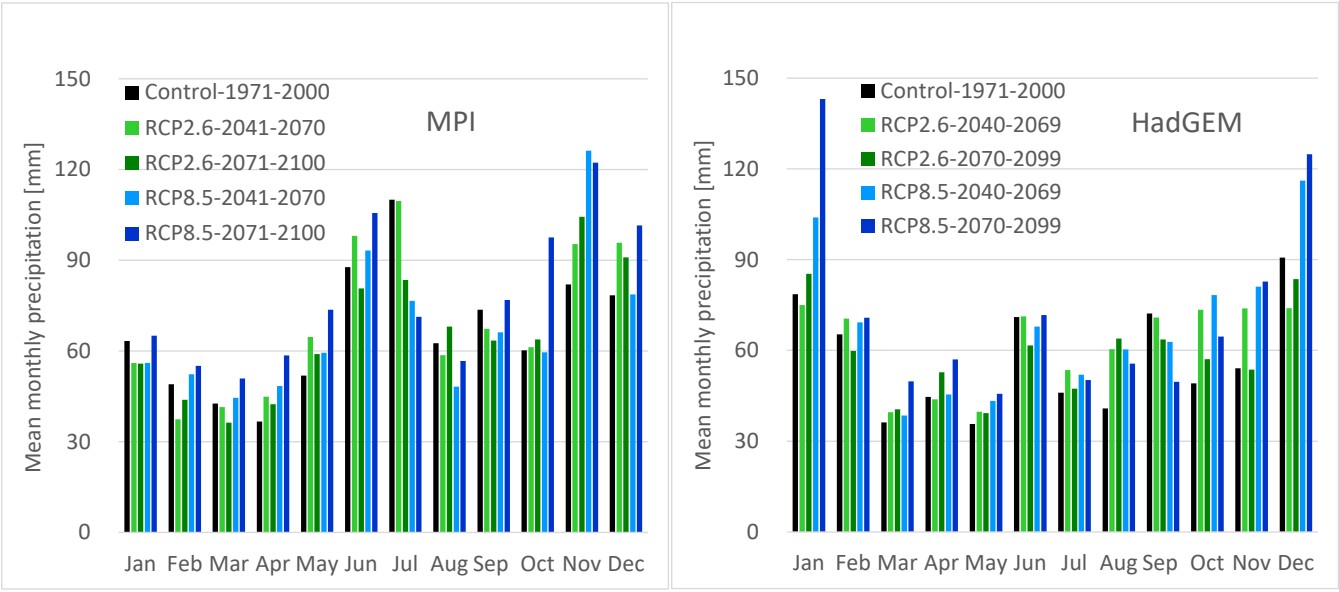


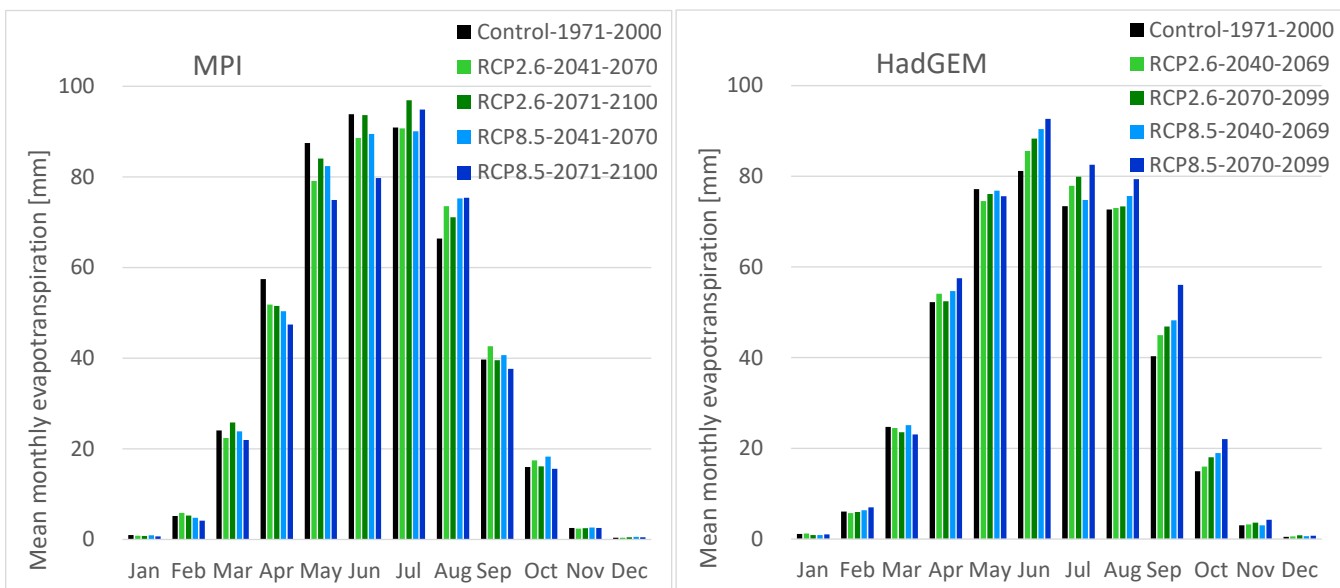

**Figure 7: Scenario and model-specific values of mean monthly precipitation (top) and mean monthly evapotranspiration (bottom).**

### 3.2.3 Impact of sea level rise on drainage capacity and frequency of system overload

The drainage capacity at tidal gauge Knock was calculated based on the simulations of the TRIM-NP model overlaid by long-term sea level rise derived from the regionalized projections of the IPCC (2021) for the German bight (Figure 8). Sea level extremes associated only with the storm events show strong inter-annual and inter-model variability with no significant trends in water level upper percentiles for the 21st century for any realization. Secular sea level rise, thus, is the main driver of substantial changes in high water levels for future scenarios. The magnitude of sea level changes is strongly dependent on the

chosen pathway scenario (here RCP2.6 and RCP8.5) with a minor influence of the driving climate model (MPI or HadGEM).

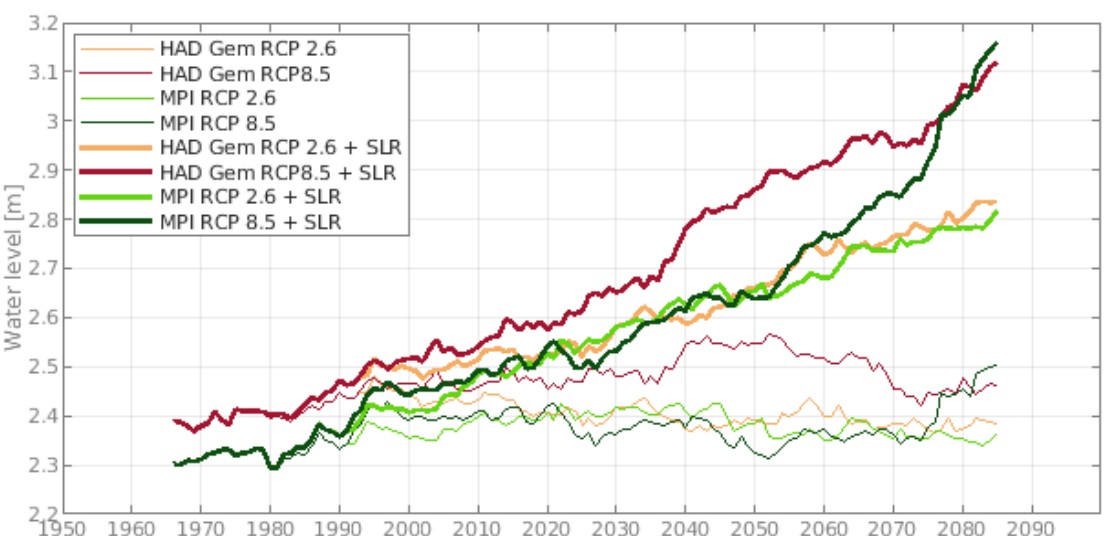

**Figure 8: 30-year running mean of annual 99.9 percentiles of sea water level at gauge station Knock (in [m] above NHN) for different projections with and without mean sea level rise (SLR), using 90 cm sea level rise for RCP 8.5, and 50 cm**

**sea level rise for RCP 2.6 by the end of the 21st century.**

    In general, high water levels disable free drainage and reduce the pump capacity, which depends on the pressure head between inland water level and sea level (Figure 3). The climate projections show a long-term decrease in drainage capacity (Figure 9), which is in agreement with the projected increase in sea level (Figure 8). While the decrease for the control period is relatively

weak (especially for the HadGEM model), it accelerates, in particular for the RCP8.5 scenario, in the second half of the 21st century, while the decrease in drainage capacity is weaker for the RCP2.6 scenario. Such a decrease can be expected to contribute to a future increase in drainage system overload even if precipitation extremes will not increase for the future.

    As expected from the projected increase in winter precipitation and the decrease in drainage capacity, the projected number of days with drainage system overload increases for all investigated combinations of climate models, emission scenarios, and

future periods. Compared to the control period, the increase is higher for the RCP8.5 than for the RCP2.6 scenario, and higher for the far future (end of the century) than for the near future. Generally, the number of days with system overload is higher in the simulations driven by the HadGEM than by the MPI model (Figure 10). This can be partially attributed to HadGEM model simulating more long-lasting rainfall events, resulting in a higher number of consecutive days with drainage system overload (compared to MPI model), while the number of separate rainfall events causing system overload is similar to the MPI model.


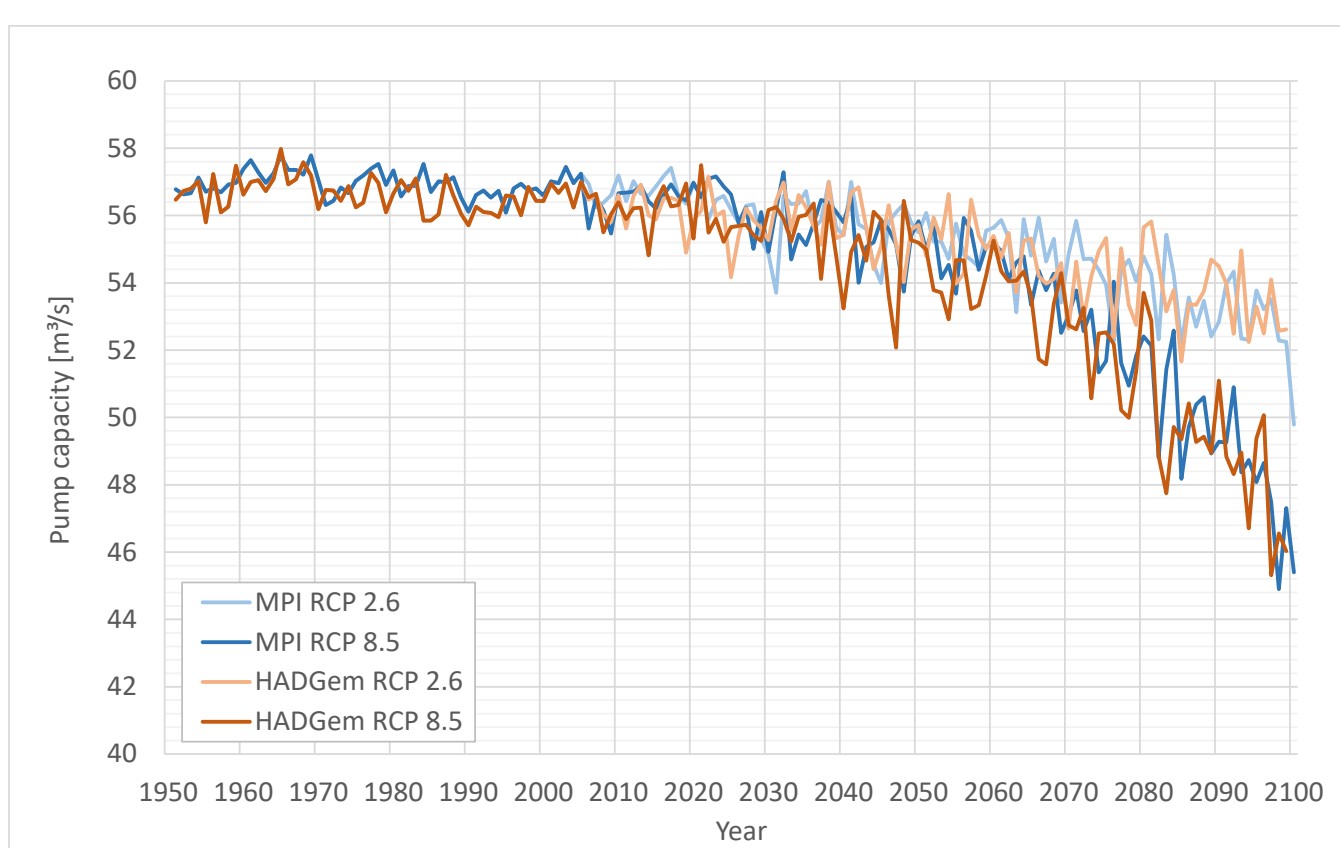

**Figure 9: Impact of sea level rise on the estimated annual pump capacity [m³/s] per pump at gauge station Knock.**

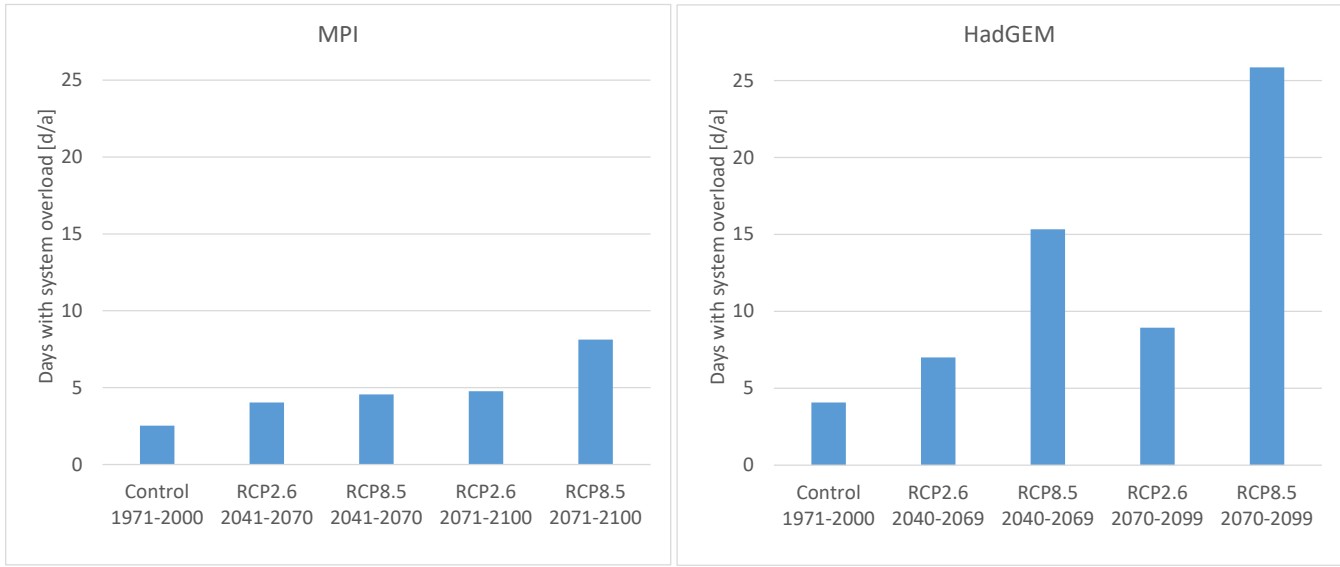

**Figure 10: Projected number of days with drainage system overload at station Knock obtained from the different models and scenarios for the mid and the end of the 21st century (MPI: left; HadGEM: right).**

### 3.2.4 Analysis of scenario calculations on compound events

As for the control period, the largest 15 winter events (October to March) of drainage system overload, sea level, and precipitation were selected from the 30-year periods centered around the mid and the end of the 21st century. The pattern of contributing drivers was analysed for a small ensemble of two climate models (HadGEM, MPI), for two concentration pathways (RCP2.6, RCP8.5), and for two different periods (near future, far future).

The projections based on the HadGEM (Figure 11) and MPI models (Figure 12) indicate that for all combinations of scenarios and models, and for both investigated periods, the highest system overloads are not generated by a combination of the extreme individual events (5-tide mean sea level, 3-day precipitation) but by compound events of moderate or intense precipitation and moderate or high coastal sea levels.

By the end of the century and for all projections, the inland flood generation mechanisms are dominated by compounding events. The general pattern of contributing drivers and the compound events is similar for the simulations driven by both climate models (Figures 11, 12) while the numbers of events attributed to the contributing drivers differ: Moderate 5-tide coastal sea levels in combination with intense precipitation generate the highest drainage system overloads. As for the control period, flood generation seems to be more sensitive to intense precipitation compared to high sea level during a storm tide. For the simulations based on both climate models, most of the high storm tides do not lead to any drainage system overload (only one event for HadGEM in the far future for RCP8.5), while most of the intense precipitation events do. Both models simulate several compound events due to a combination of moderate drivers: HadGEM in particular, since simulated system overloads

are almost only simulated for moderate storm tides (see also Table 1). The magnitude of the driving precipitation events and their timing also differ between the climate models (Figures 11, 12). While for the MPI model the antecedent 3-day precipitation is higher (mainly generated by rainfall events in late autumn), antecedent 3-day precipitation of the HadGEM model is smaller (mainly generated by precipitation events in winter). In both climate models and scenarios there is a high inter-annual variability in storm activity. The seasonal distribution of storm surges is not expected to shift significantly towards the end of the century (Figure 13). A shift in the timing of the events could result in a decreasing likelihood that storm tides and intense precipitation coincide.

As indicated by the development of contributing drivers, especially for the pessimistic representative concentration pathway RCP8.5, compound events probably become more intense and more frequent until the end of the 21st century. The magnitude of drainage system overload will become significantly larger with intensification of the single drivers (concerning the max. system overload: by a factor of 5 for the HadGEM model, and by a factor of 3.5 for the MPI model). Moderate events contributing to the compound events will also be more intense in the future. For the RCP8.5, drainage system overload is expected to be significantly larger by the end of the century compared to the near future. Analysing the projections for the more optimistic emission scenario RCP2.6, the results indicate that the generation mechanisms of compound events remain constant, as well, while the magnitude of system overload does not increase from near to far future for both climate models. Drainage system overload within RCP2.6 simulations seems to be mainly caused by intense rainfall situations.

Similar to the observations and the control period, assuming a multivariate compound event does not entirely explain the pattern of the events, indicating that preconditions such as antecedent soil moisture could provide better insight.

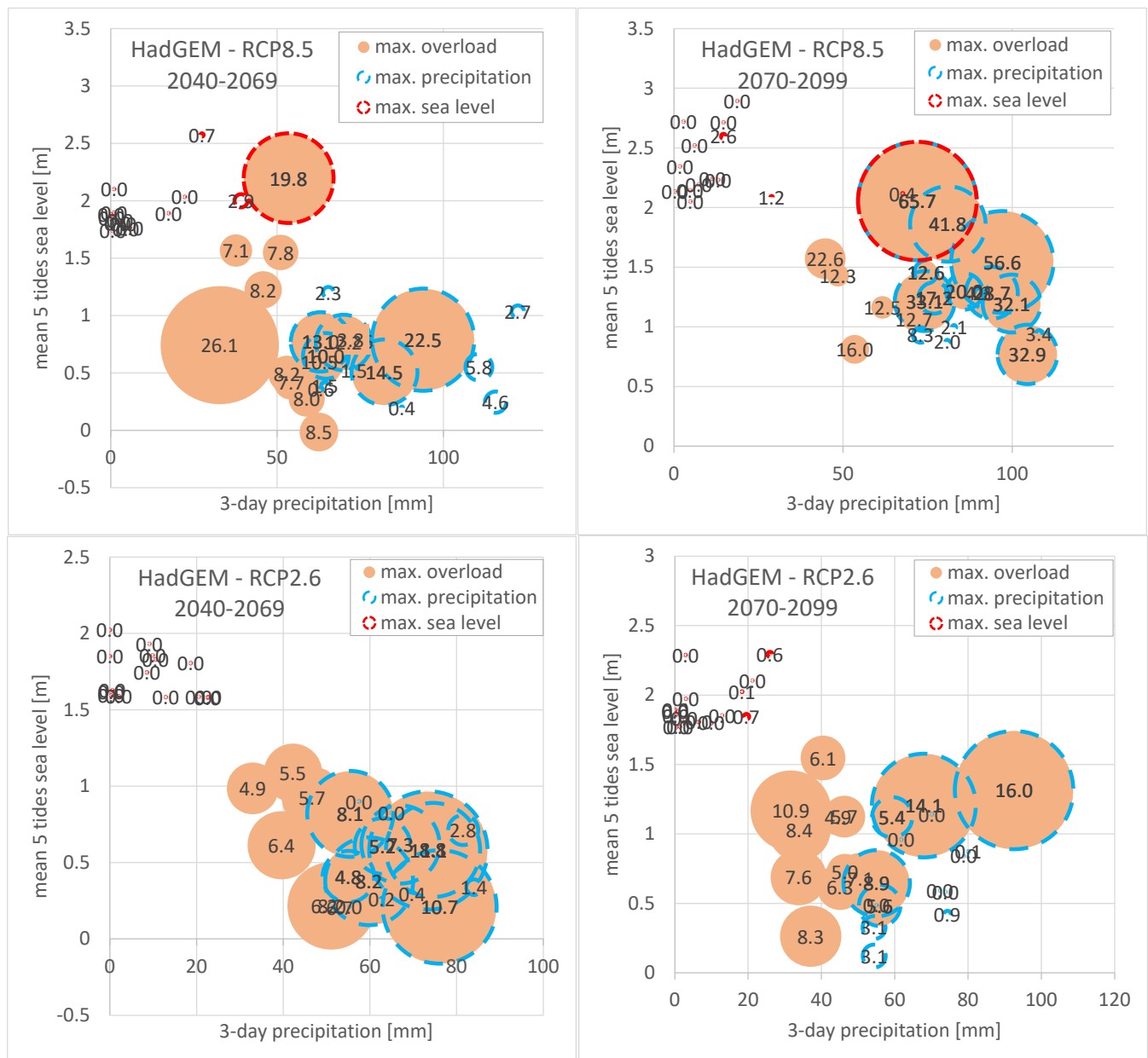

**Figure 11: Simulated drainage system overload [10⁶ m³] for the 15 highest sea level (mean sea level over a 5 tides period above NHN [m]; red circles) and precipitation (antecedent 3 days precipitation sum [mm]; blue circles) events in the winter half-year (October to March) in the period 2040-2069 (left) and 2070-2099 (right) for the RCP8.5 (top) and RCP2.6 (bottom) scenario derived from the simulations driven by the HadGEM model data. Additionally, the 15 simulated drainage system overloads in the same period are shown (orange circles). The diameter of the circles is proportional to the magnitude of the modelled drainage system overload (number in the circles/bubbles in [million m³]).**

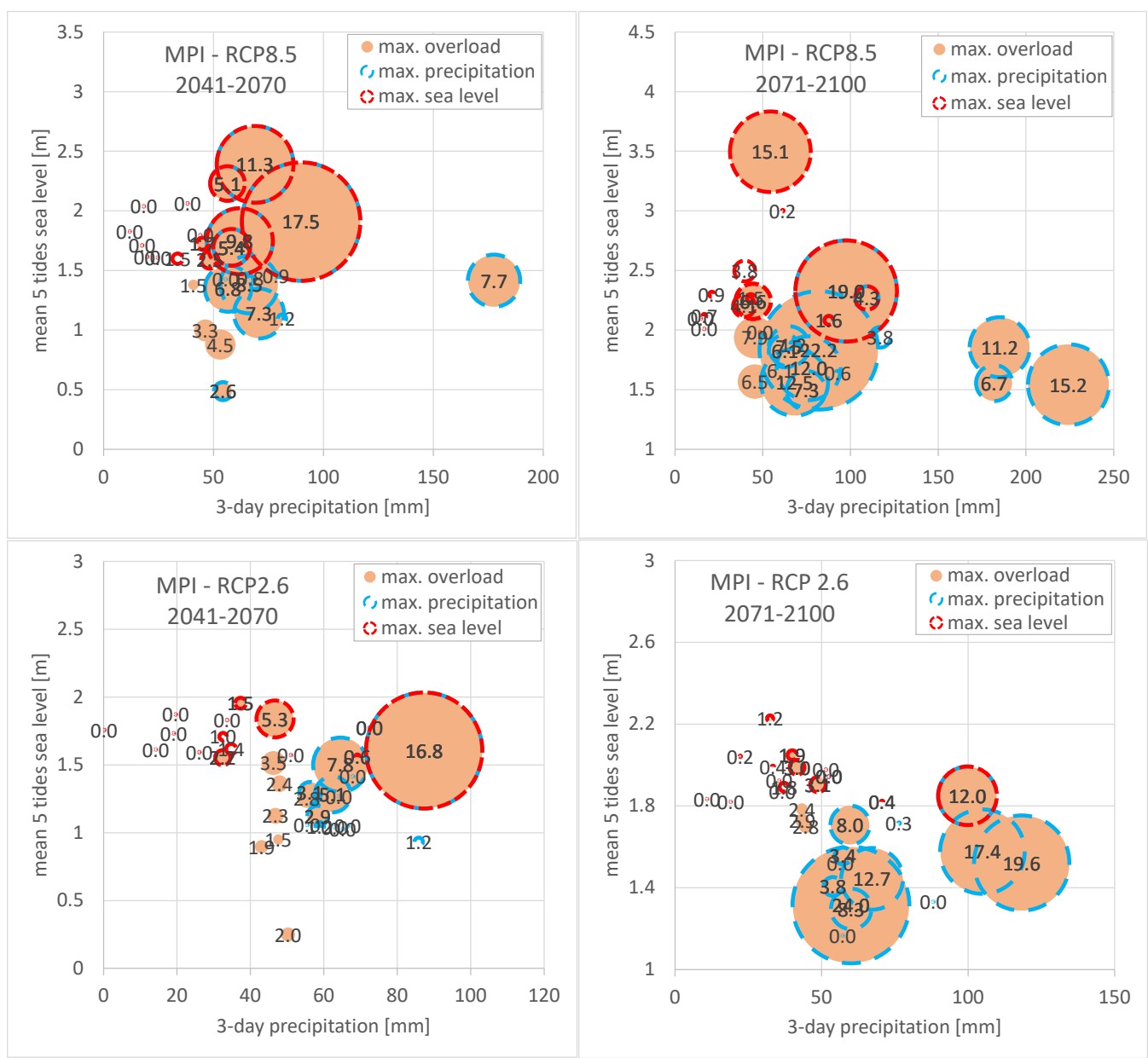

**Figure 12: As Figure 10 but for the simulations driven by the MPI model. Note that periods for the analyses are shifted by one year compared to Figure 10 because of the shorter duration of the HadGEM simulation (number in the circles/bubbles in [million m³]).**


**Table 1: Frequency of the co-occurrence of the individual drivers (storm tides and precipitation) related to their intensity, for the largest 15 events of drainage system overload. SL = sea level; PREC = precipitation. For definition of extreme, high/intense, moderate events see section 2.3.**

| Data / Climate model | Emission scenario | Time frame | No of events: high SL – intense PREC | No of events: high SL – moderate PREC | No of events: moderate SL – intense PREC | No of events: moderate SL – moderate PREC |
|---|---|---|---|---|---|---|
| Data | | 2000-2019 | 3 | 1 | 5 | 6 |
| MPI | | Control | 4 | 2 | 6 | 3 |
| HadGEM | | Control | 0 | 0 | 8 | 7 |
| MPI | RCP8.5 | Near future | 5 | 2 | 5 | 3 |
| | | Far future | 2 | 2 | 7 | 4 |
| | RCP2.6 | Near future | 1 | 3 | 4 | 7 |
| | | Far future | 1 | 3 | 8 | 3 |
| HadGEM | RCP8.5 | Near future | 0 | 1 | 5 | 9 |
| | | Far future | 1 | 0 | 9 | 5 |
| | RCP2.6 | Near future | 0 | 0 | 8 | 7 |
| | | Far future | 0 | 0 | 3 | 12 |


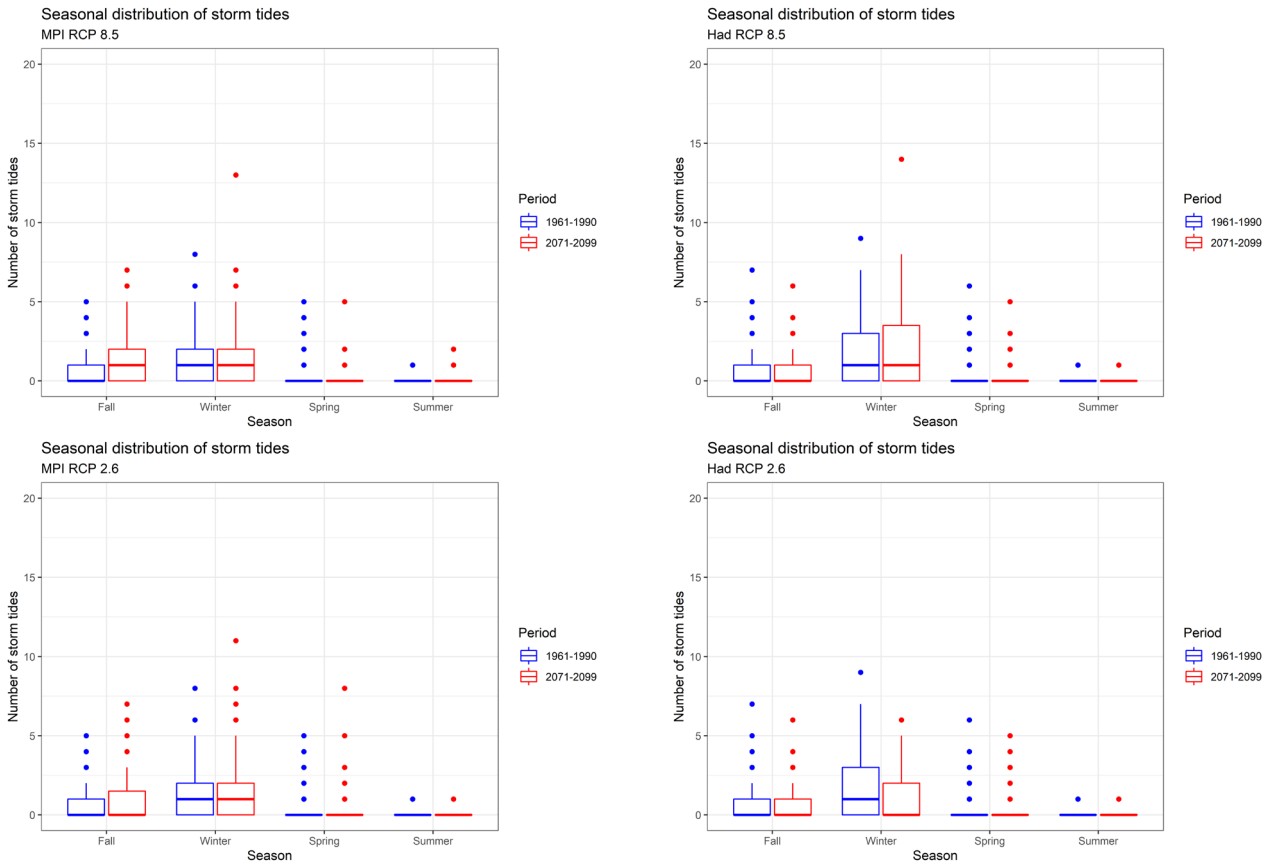

**Figure 13: Seasonal frequency distribution of storm tides in the analysed scenarios. RCP8.5 (top), RCP 2.6 (bottom), MPI (left), HadGEM (right).**

## 4 Discussion

The analysis of historical data revealed that compound events of moderate storm tides and intense rainfall caused inland coastal flooding due to an overload of the drainage system of the Emden water board, rather than individual drivers. This is in agreement with observations and analyses from the Dutch coast (van Hurk et al., 2015), from China (Xu et al., 2023), and from the UK (Svensson and Jones, 2002), although compound events of storm tides and high river discharge are not observed more frequently for the Ems estuary than to be expected by chance (Heinrich et al., 2023). Since all these sites are located at shallow
coasts with tidal influence and are more or less facing the main wind direction, we assume that findings are spatially transferable to coasts with similar characteristics.

Heinrich et al. (2023) found that only rivers along the westward-facing coasts of Europe experienced an increased probability of simultaneous storm tides and high river discharges. For the area of coastal water boards, the exposition of the coastline seems to be of minor importance as long as they are exposed to the west or northwest, since the location of water board areas usually is close to the dike (to the North Sea), compared to the headwaters of the rivers flowing into the North Sea. Therefore, we assume that the findings of our study are transferable to large parts of the Dutch, German, and Danish low-lying North Sea coast. Based on a comprehensive ensemble study Heinrich et al. (2024) showed that in all climate models and scenarios, an increased frequency of atmospheric cyclonic westerlies in winter is to be expected by 2100. An increasing flood risk can be assumed for westerly estuaries while not for northward estuaries like the Ems River. Heinrich et al. (2023b) further showed that in future scenarios compound events will increase mainly because of rising mean sea level which will further increase the risk and occurrence of events. For water boards located upstream in an estuary, high river discharges will become more relevant. Even if Heinrich et al. (2023) did not reveal an increased probability of a coincidence of storm tides and high river discharges, such events nevertheless may induce increased water levels in the river.

Individual projections on climate change's impact on winter precipitation (Figure 7), runoff generation (Bormann and Kebschull, 2023), and sea level rise (IPCC, 2021) suggested that drainage of coastal regions might become a greater challenge for the future (see also Spiekermann et al., 2023). The question arose whether, as observed in the past, compound events can be assumed to cause the largest system overloads (inland flood events) also in the future, or whether the importance of contributing drivers might change. The simulation results of the modelling experiment emphasize that the generation mechanisms and the resulting patterns of current inland flooding at the German North Sea coast can be simulated at the water board scale by the model set-up applied in this study. This is essential for the predictability of future compound events under climate change conditions, needed for the adaptation to long-term climate change impacts. The model projections indicate that the flood generation mechanisms remain stable for future time periods under climate change conditions. The results based on the small ensemble also suggest that in accordance with Seneviratne (2012), compound events of moderate individual drivers will be an important source of inland flood risk until the end of the 21st century. The simulation results based on both climate models are consistent concerning runoff generation mechanisms despite model-specific differences in the projections (e.g., precipitation projections, Figure 7; contribution of high storm tides, Figures 11, 12), indicating that such behaviour might also be confirmed by larger model ensembles. However, this needs to be confirmed by future studies as soon as larger climate ensembles are available in high temporal resolution.

The magnitude of drainage system overloads is expected to increase significantly for the RCP8.5 scenario by the end of the century, while intensity for system overload seems to remain constant for the projections based on the more optimistic representative concentration pathway RCP2.6. This is in accordance with other climate change impact studies on hydrological systems in Germany (Brasseur et al., 2023). The intensity of precipitation events seems to be the more important driver compared to the average sea level during one or a series of storm tides. Nevertheless, as long as rainfall amounts and intensities will not exceed those represented by the climate projections, moderate rainfall alone - without any restriction in drainage

capacity - is not expected to lead to large floodings in shallow coastal inland regions along the North Sea coast until the end of the 21st century.

The projections emphasize that climate change adaptation will be required also for all scenarios considered. In accordance with Cioffi et al. (2018), existing drainage infrastructure will not be sufficient to cope with the consequences of climate change, even for more optimistic emission scenarios (RCP2.6). As shown by this study, adaptation thereby does not only need to consider the individual impacts of sea level rise (Bormann et al., 2023; NLWKN, 2007) and runoff generation (Bormann and Kebschull, 2023), but explicitly the potential impacts of compound events, because highest drainage system overload must be expected by such type of combined events.

Despite the representation of past events, the study suffers from some uncertainties. First, the ensemble of climate models and climate change scenarios is small. It remains possible that combinations of other climate models and emission scenarios lead to a different climate change signal. We already recognized that projections and process representations of MPI and HadGEM models differ. This needs to be reviewed using a larger ensemble as soon as more long-term climate model simulations are available that provide forcing data for the impact models in at least hourly resolution. Second, the storage-based representation of the water spatially distributed in the water board area neglects internal flow processes. Better data on such internal flow processes and state variables could reduce this uncertainty. Nevertheless, as long as the management of the anthropogenic drainage system does not follow clear and systematic rules, and is influenced by unknown or even erratic individual decisions, a process-based simulation of the drainage system will probably fail. Third, while for precipitation and temperature, a bias correction of the climate scenario simulations was required (input for SIMULAT), no bias correction was carried out for the wind speed (input for TRIM-NP). This might cause some inconsistency concerning the driving forces of this study which could be quantified by model intercomparison with/without systematic bias correction. Finally, the observed and simulated data show that a combination of intense individual events (one or more storm tides and rainfall events) must not necessarily cause the largest system overloads in the simulations. Probably there are still other influencing factors (such as a high soil saturation of the water board area, e.g. caused by a longer rainfall memory), which are not yet considered by the approach presented above. Thus, assuming a multivariate compound event based on storm tides and intense rainfall does not entirely explain the pattern of the events, additional modelling studies could reveal its impact, since usually soil moisture observations are not available. Concerning the typology of compound events, we agree with Zscheischler et al. (2020) that "the complex nature of compound events results in some cases inevitably fitting into more than one class, necessitating soft boundaries within the typology". For flood generation at shallow coasts, we suggest explicitly considering combinations of the defined types (e.g. multivariate and preconditioned events) to represent the complex regional flood generation process.

## 5 Conclusions

The results of this study indicate that the highest overload of coastal drainage systems currently is and in the future will be caused by compound events of moderate to intense storm tides and intense rainfall. While intense rainfall, in particular, poses

a big challenge for coastal drainage systems, the simultaneous occurrence of storm tides and rainfall will pose major problems for the defence systems. Therefore, the dimensioning of coastal and flood protection measures as well as the related risk management explicitly must take such events into account to avoid flood events and resulting damages.

However, such type of compound events is considered explicitly neither in current risk management nor in long-term planning of climate change adaptation in Germany. So there is a need to rethink the question of to what kind of events and hazards a society needs to adapt to. While this study can contribute to emphasize that compound events need to be considered for adaptation planning and risk management, for a quantitative assessment larger climate model ensembles in high temporal resolution need to be considered in future studies.


*Data availability:* The data used in this paper are available from the authors upon request.

*Author contributions*: The conceptualization of the study was carried out by HB and RW. The simulations were performed by JK and LG. The evaluation of observations and simulations was carried out by HB. All authors contributed to the preparation
and the revision of the manuscript.

*Competing interest*: The authors declare that they have no conflict of interest.

*Financial support*: This investigation was funded by the German Federal Ministry of Education and Research (BMBF) (grant
numbers: 01LR2003D, 01LR2003D1, 01LR2003A, 01LR2003A1)

**Acknowledgement**

This investigation was funded by the German Federal Ministry of Education and Research (BMBF) (grant numbers: 01LR2003D, 01LR2003D1, 01LR2003A, and 01LR2003A1) within the project WAKOS – Wasser an den Küsten
Ostfrieslands (RegIKlim program).

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
