# Peer review of "Model-based assessment of climate change impact on inland flood risk at the German North Sea coast caused by compounding storm tide and precipitation events"

_EGUsphere, 2024_

## Author Response (AR1)

Dear reviewers,

Thanks a lot for your helpful comments which improved the clarity our manuscript. We go through your comments point by point to clarify how we will use your comments and questions during revision.

**Reviewer 1:**

**General comments**

1. This is an interesting paper showing that compound flooding at the Northwestern coast in Germany, can be driven by combination of moderate or intense drivers and not only by extreme individual drivers now as well as under future climate projections. The method is clear and appropriate. The results are relevant for the scientific community working on compound flooding as well as for adaptation and mitigation planning of local and regional water boards of the studied area. Below comments indicate elements of the manuscript that are not yet (entirely) clear to me.

   *Thanks a lot, we appreciate your positive assessment of our study.*

**Specific comments**

1. Title: Could be more concrete mentioning the study area instead of "coastal areas" as I think the results are mainly relevant for the particular case study rather than for coastal areas in general.

   *Thanks for the comment, we adjusted the title to: "Model-based assessment of climate change impact on inland flood risk at the German North Sea coast caused by compounding storm tide and precipitation events"*

2. The authors use the term "storm tide", "storm surge", "sea level", "maximum sea level" and "storm flood" throughout the text. Are these used interchangeably or with different meanings? A common definition is water level = tide + surge (+ sea level rise), but it does look like the authors use them to refer to the same. An introduction and definition of the terms used would provide more clarity. I also suggest to use one term throughout for clarity.

   *You are right, suggesting standard terms for the description of the extreme events we analyse. Therefore, in the original manuscript, we used the terms according to Pugh and Woodworth (2014): Sea Level Science: Understanding Tides, Surges, Tsunamis and Mean Sea-Level Changes (https://doi.org/10.1017/CBO9781139235778) without citing the paper explicitly for the definitions used.*
   *In the revised version of the text, we explain the term used (storm tide) in section 2.3 and reduce the use of different terms as suggested (the terms storm flood and storm surge were removed, while we still use the term sea level).*

3. The authors focus on events of storm tide and precipitation events. Why do they not consider river discharge as well? It seems the authors see river discharge and precipitation as strongly linked in the area of the study (e.g., by mentioning the work of Henrich et. 2023 on storm tide and river discharge), but this is not made explicit. Could this be clarified and elaborated?

*This is a good point. The Emden water board is located downstream the Ems River, at the Ems estuary. At gauge Knock, the estuary is more than 3 km wide, so the impact of river discharge on the water level in the estuary is small compared to the impact of storm surges. Furthermore, there is a lag between high river runoff and precipitation depending on catchment size while due to the small area in this study surge and precipitation events occur simultaneously. So we can not rule out that there is a contribution, but a different time lag is needed for the largest impact. Therefore, we neglected such influence. However, for the transferability of our results e.g. to upstream water boards at the Ems River, this is essential. We added information on the estuary and the catchment size of the Ems River to section 2.1 (target area) and addressed this issue in the discussion section (also requested by reviewer 3).*

4. 10 I think the abstract could be clearer when describing the methods and the results separately (e.g. "We analyzed historical data and simulations of a control period as well as a future period. The results are X". Instead of "Our analysis of historical data showed X. Our analysis of simulation data for the control period also showed X.") as well as more concrete (e.g. "combinations of climate model A and climate model B with emission scenarios RCP8.5 and RCP2.6" "combinations of two highly resolved climate models and two emission scenarios")

   *We rewrote the abstract while considering your suggestions.*

5. 23-27: Looking at the references, this seems to describe the case of the North Western Coast of Germany and should be mentioned somewhere.

   *We added further literature from other countries and continents in the first sentence of the introduction, and we added the regional focus of our study as suggested to the following sentences.*

6. 63 what is meant by "integrative scenario-based climate impact assessments"? Is this a standard term?

   *We did not aim at using this expression as a standard term. We used it to express the integrative calculation of the potential impacts of storm surges and rainfall based on a consistent set of climate scenarios. We rephrased this paragraph.*

7. 64, the authors state that such "integrative scenario-based climate impact assessments" are not available for local to regional flooding at shallow coasts. Could the authors provide references for global analyses or non-shallow coasts? Or similar efforts focusing only on historical data, but not for future climate projections? I am missing a more comprehensive literature review here on studies related to compound flooding, and in particular approaches using physics-based models.

   *You make a good point. As mentioned in the manuscript, we did not find any comparable modelling study in the literature. On the global scale, water management along shallow coast is not explicitly considered. However, data based studies are available (e.g., by van den Hurk et al., 2015), and thankfully you provided the citation of Xu et al. (2023) in your review, which shows similarities coupling an urban storm water model to SLR scenarios. We extended this*

*part of the literature review in the revision as suggested (also requested by the other two reviewers).*

8.  120 Different types of compound events are introduced as part of the method section, but not used or reflected upon in the remainder of the study. If there is no explicit link with the results or discussion sections, this should be left out.

    *Thanks for this note. We used the classification only implicitly for the discussion of the explanatory power of our model approach. In the modelling approach presented, we assume that the multivariate mechanism can explain the pattern of the events (information added to section 4). But this is not sufficient, preconditions such as soil moisture in the marsh are affects the pattern significantly. We therefore also revised the manuscript in order to use the classification explicitly in the discussion of the identified compound events in the results and discussion sections).*

9.  135: An explanation (summary from van den Hurk) of the reasoning behind selecting the 15 largest events would be helpful.

    *Van den Hurk et al. (2015) showed the 20 largest inland water levels and compared them to the highest 10 wind-induced surges and precipitation events. Due to limited data availability (20 year time series of observations) we made compromise between the relative small number system overloads in the observations and a minimum number of events for evaluation. We ended up in 15 events, which is in the same range as applied by van den Hurk et al. (2015), and which we applied to observed water levels, wind-induced surges and precipitation events. We added this information to section 2.3.*

10. 137. "by selecting the 15 largest events according to sea level, precipitation and inland water level". I was not sure if the authors meant 15 events in total or 45 events in total until I saw Figure 5.

    *We selected 15 events each, resulting in max. 45 events if the selected events were independent. We clarified that in section 3.1 during revision.*

11. 142 I was confused by the sentence "While the analysis [...] for the evaluation of the climate change projections a new model set-up was necessary." In the end, the authors still follow the approach of van den Hurk et al. 2015, but instead of historical data they use simulated data for which they had to set up a model. Is this what is meant here?

    *Yes, this is one aspect. To be able to analyse future conditions, we use simulated data instead of observed data. Another – more important – aspect is that we needed to replace the observed water level [m] by the simulated system overload [m³]. This is due to the fact that we are no able to model the whole anthropogenic drainage system by a hydrodynamic model. This is because the local water management action is not documented. Therefore, we compare runoff generation against drainage capacity, resulting in a system overload described by a water volume, instead of calculating inland water levels. This information has been added to section 2.3.*

12. Figure 2: This flow-chart is helpful to understand the model chain. I suggest to expand it to include the entire methodology of the paper.

*We think that one flow chart including the whole methodology is too complex. We assume that by adding additional information in the sections 2.1 to 2.3 the methodology becomes clear now.*

13. Authors describe magnitudes with terms as "largest", "moderate" or "intense" and "heavy", but it remains unclear what these mean.

    *This is a good point, also mentioned by the other reviewers. We defined the terms "extreme" , "intens/high" and "moderate" in section 2.3 and reduced the number additional terms as far as possible.*

14. 210 / Figure 3: Are the circles the underlying data points and the line the fit? Could the authors provide more context and how this function is obtained? What has been received? Can this data be shown? How was the polynomial estimated?

    *As mentioned in the paragraph prior to Figure 3, the drainage function of the pumps at gauge Knock is composed through an assessment of the state agency for the free flow through the gate in case of low tide (=negative hydraulic head) and a calculation of the pump capacity per pump, based on the pump characteristics and of the blade position (positive hydraulic head). According to the Emden water board, for every 50 cm (= for each 50 cm interval of hydraulic head) the blade position is adjusted, and therefore a new estimation of pump capacity is necessary using the pump curve (points). Above a hydraulic head of 3.75 m, the pumps need to be switched off. The polygon is the best fit to the pump rate estimations, selected by least squares method.*
    *We don't think that providing scans of pump characteristics (e.g., through an additional figure), and the blade positions for every 0.5 m is beneficial for the readers. But we added the information provided above to section 2.4.3 and to the figure caption.*

15. 220-225: What do "very high", "high" and "moderate" mean? Could this be made more concrete in the results and discussion section? The same holds for adjectives "intense" and "heavy" that the authors use in other parts of the results and discussion section.

    *We added a definition of extreme, intense/high and moderate events to section 2.3 and removed unnecessary additional adjectives.*

16. 223 "… lead to the largest observed deviations from the regulated inland water level (in total 9 of the 15 events)": Which 9 events are these? This is not clear from Figure 4. And where and why has this threshold in deviation from regulated inland water level been set?  How is the size of the circle related to the difference? Linearly? Logarithmic? Would the scale impact how we interpret the results? In general, the interpretation of the Figure is not immediately intuitive to me. Perhaps the authors could guide the reader through the three groups (of 15 events each) separately and then arrive at their overall conclusion.

    *The 9 events you ask for are shown in Figure 4. According to the figure caption, those events are the orange circles with red or/and blue edging. The remaining 6 orange circles (without any red or blue edging) are those compound events which member belong neither to the 15 highest sea levels nor to the highest 15 precipitation events. As mentioned in the caption, the*

*diameter of the circles is proportional to the values. As suggested, we added the description to the text of section 3.1.*

17. Figure 4: The caption states that the Figure shows the water level anomaly [cm] however the number in the circles run from 1- 45. These are probably event IDs and should be changed to water level anomaly values?

    *This is not correct; the number are the water level anomalies in [cm] and not IDs. The diameter of the circles is proportional to the values. We added this information to the figure caption.*

18. 247 Why is the control period not chosen to be identical with the period for which observational data is available? Wouldn't this make the comparison more straight forward and, for example, enable plotting using the same axes limits?

    *We chose a standard control period, which is usually used for the interpretation of climate model output. The time period of the observed data (20 years) is too short to use it as control period for climate variables (>= 30 years).*

19. 321 "most of the storm tides do not lead to significant drainage overload": How is "significant" defined? Why?

    *The term significant was not used in a statistical sense. From the figures 11 and 12 it can be seen that that most of the highest storm tides do not induce any drainage systems overload. We adjusted the wording to make that clear.*

20. Figure 10 should be plotted in the same Figure or at least with the same y-axis. Now the results look more similar than they are.

    *It's correct that the results look more similar than they are. We adjusted the axes. However, the general trend (increasing number of days with system overload at the end of the century, higher number of days with system overload in the RCP8.5 scenario) is similar for both climate models.*

21. In some Figures HadGEM is on the right panel and in others on the left. I suggest to keep this the same.

    *Thanks for this note. We adjusted the order of the subfigures.*

22. Figure 11: There seems to be a negative correlation between sea level and precipitation. How could this be explained and how is this of relevance to the results?

    *This is a good question. However, you should keep in mind that the correlation might be an artefact by putting the three separate ensembles (15 events each for highest sea levels, highest drainage system overloads, highest precipitation events) together. Within the three ensembles, there is no such trend visible. Anyhow, one explanation could be that the highest storm tides are generated by NW winds, while the highest precipitation is generated by westerly winds (causing only moderate storm tides).*

23. 351: The authors cite Heinrich et al who states that compound events of storm tide and high river discharge are not observed more frequently than to be expected by chance. How will this be in the future? What are the limitations to using individual projects for winter precipitation, runoff generation and sea level rise (l. 357)

*Heinrich et al. (2023) showed that the simultaneous occurrence of storm surges and storm surge series in conjunction with long lasting precipitation predominantly occurs during cyclonic westerly weather conditions. Therefore, westward estuaries are mainly at risk. Based on a comprehensive ensemble study it could be shown that in all climate models and scenarios an increased frequency of atmospheric cyclonic westerlies in winter is to be expected by 2100 (Heinrich et al., 2024). So, an increasing flood risk can be assumed for westerly estuaries while not for northward estuaries like the Ems River.*
*We will comment on this in the revision. Heinrich et al. (2023b) (10.3389/fclim.2023.1227613) further showed that in future scenarios compound events will increase mainly because of rising mean sea level which will further increase the risk and occurrence of events. We included these issues in the discussion.*

24. 411: It would be great if the data could be published alongside the paper.

*We will discuss this option. First, data will be made available on demand.*

25. This seems to be a similar paper focusing on a different case study: https://www.sciencedirect.com/science/article/abs/pii/S0022169423001087

*Thanks for this note! We considered this paper for the literature review and for the discussion.*

**Reviewer 2:**

**General comments**

2. This manuscript evaluates the role of storm tides and precipitation and their compounding effects in creating drainage system overloads for a case study are in Germany both under present-day and future climate conditions. This is an interesting and timely topic and the manuscript will be a valuable addition to the existing literature. I provide some general and several detailed comments below which should be taken into consideration before the manuscript can move toward publication.

*Thanks a lot, we appreciate your positive assessment of our study.*

3. I would put less emphasis on the differences in changes found between the two emission scenarios (for example in the abstract). The fact that 8.5 leads to more change than 2.6 is to be expected. Personally I find the results for present-day most interesting showing which types of events cause the most problems, and the fact that models can at least capture the broad characteristics of those events. The future change part is less exciting I would say, so I suggest not putting it too much at the forefront.

*Thanks for sharing your opinion on which parts of our investigation are more interesting than others. Indeed, the present-day analysis reveals which kind of events cause most water management problems, and the results are in agreement to those found for the Netherlands*

*by van den Hurk et al. (2015). Reviewer 1 made some suggestions to make the main findings intuitively understandable. In addition, we linked our case study to the classification according to Zscheischler et al. and discuss the necessity to identify further precondition in addition to the assumption of a multivariate compound event type te better explain the pattern of events.*

*Unfortunately, data availability of German coastal water boards is scarce so that a spatial comparison for different locations is not possible. In our opinion it seems to be crucial to test whether model based analysis can depict the impact of compound events on coastal flood risk and on the potential system overload of drainage systems. Therefore, we think that the modelling study has a significant added value and innovation aspect, as also emphasized by reviewer 3. Applying the model chain to the control period proves that the generation mechanism of compound events is represented. Its application to future conditions confirms that compound events will probably also happen in the future and will be more intense than under today's conditions. However, we agree with you, that it's not necessary to discuss the differences in the scenarios in too much detail. We tried to focus the discussion in the revision on the ability of the models to reproduce the generation of compounds and their drivers, and not to discuss the differences in the scenarios in too much detail, without affecting the main findings. In general, it seems to be relevant to consider the generation of hydro-meteorological compound events for climate change adaptation planning.*

4. The referencing is very biased toward self-citations (some references only available in German), and while that is in general ok and often warranted, there are parts where other literature should be cited; for example, in the intro when the broader topic of compound flooding is introduced, but other places as well.

   *Thanks for reminding us to put more emphasis on considering the available literature. In particular, in the introduction we replaced some of the the self-citations by citing other relevant publications to describe the challenges and the status quo of investigations and available solutions. This helped also to relate our regional challenge to similar coastal regions. However, we did not find suitable other references for all statements. We extended the literature review, also thanks to reviewer 1 suggesting another similar study to discuss.*

**Specific comments**

26. 11 storm surge series?

    *Thanks for this correction.*

27. 25/26 are the "storm floods" from inland or coastal flooding drivers? Terminology is not clear in many places when referring to different flood drivers.

    *Thanks, your right. Here, storm tides are meant, as well. We critically checked the wording throughout the manuscript during revision.*

28. 36 typo: the end of the century

    *We corrected the typo.*

29. 48/49 I couldn't follow that sentence. The fact that drainage systems worked at the edge of capacity means they weren't overloaded, but the occurrence of regional flooding indicates otherwise.

    *Thanks for this note. In general, regional water managers report that the drainage systems more often work at the edge of their capacity in our days. But during the described event in 2022 local flooding occurred, because of a system overload.*
    *We rewrote the sentence accordingly.*

30. 64 see above, what does "storm flood" mean here

    *Storm flood means storm tides, here.*

31. 68 The models should not just capture tidal water level dynamics but water levels in general including storm surge

    *You are correct. We rewrote the sentence according to your suggestion.*

32. 97 typo: of sea water level

    *We corrected the typo.*

33. 108 typo: are part of the  CMIP5

    *We corrected the typo.*

34. 120-130 Zscheischler et al. (2020) do not provide a definition of compound flooding but a typology; an alternative definition to the one from the IPCC was proposed in Zscheischler et al. (2018; https://doi.org/10.1038/s41558-018-0156-3)

    *Thanks for this suggestion! We considered your suggestion and rearranged the sentence during revision.*

35. 163 typo: for potential evapotranspiration

    *We corrected the typo.*

36. 230 Figure 4 and other figures as well, please make clear what the numbers in the circles represent exactly

    *In Figure 4, the numbers in the circles are the observed deviations from the target inland level [cm], as mention in the figure caption.*
    *In Figure 5 the numbers quantify the calculated drainage system overload [$10^6$ m³], as mention in the figure caption.*
    *To be able to analyse future conditions, we use simulated data instead of observed data (see explanation in section 2.3, shortly repeated in section 3.2.1). Here we needed to replace the observed water level [cm] by the simulated system overload [m³]. This is due to the fact that we are no able to model the whole anthropogenic drainage system by a hydrodynamic model (see section 2.3 for explanation). This is because the local water management action is not*

*documented. Therefore, we compare runoff generation against drainage capacity, resulting in a system overload described by a water volume, instead of calculating inland water levels. We revised the text to avoid any misunderstanding.*

37. 300/301 Why do you think that is? It would help the reader to get some idea about where such differences could come from.

    *As discussed in a later section of the manuscript, the two climate models simulate extreme most of the extreme events in different seasons, MPI in late autumn, HadGEM in winter. Also the characteristics of the simulated events differ. The HadGEM model simulated long-lasting events, resulting in a higher amount of days with system overload (compared to MPI model), while the number of critical events is similar to the MPI model.*
    *We included this information into the revised manuscript (sections 3.2.3 and 3.2.4).*

**Reviewer 3:**

**General comments**

0.  The manuscript "Model based assessment of climate change impact on inland flood risk in coastal areas caused by compounding storm tide and precipitation events" shows how different combinations of sea levels and precipitation at present and in the future combine to drive inland flooding events/drainage system overloads at a water board in Germany. The manuscript is well written, and the conclusions seem mostly backed up by the results. I have a few comments, summarized below, that I think would strengthen the results presented in the manuscript.

    *Thanks a lot, we appreciate your positive assessment of our study.*

1.  One of the main conclusions of the research is that the combination of moderate storm tide and precipitation events drive inland flooding or drainage system overload. However, what is considered "extreme" versus "moderate" across the distribution of events, is never sufficiently defined. The 15 highest events end up being the metric that is designated as "extreme", so my assumption is anything below those events are considered low or moderate. Looking at Figure 4, the 15 highest sea levels (defined as the average across 5 tidal cycles) for the observations are all above 0.8m and the 15 highest 3-day precipitation events are all above 35-40 mm. This would lead me to believe any combination of sea level below 0.8m or precipitation below 35-40 mm would be "moderate" or "low." However, when moving to the drainage capacity and future climate scenarios for the winter, "extreme" sea level increases to over 1.5 m or 2.0 m, depending on time period and climate forcing, and precipitation extremes increase to over 50 mm, but even as high as 60 or 70 mm. Thus, in the future scenario, there are many combined events that fall over what was called the "extreme" range in the observations, even if they are not the most extreme in the new climate, which leads me to conclude that extreme sea level and precipitation events are causing many of the drainage capacity extremes, rather than moderate events. So, a broader discussion on the definition of extreme vs moderate events is necessary for backing up the main conclusions of the work.

*This is a good point, also mentioned by the other reviewers. We defined the different expressions ("extreme", "intense/high" and "moderate") in section 2.3 and reduced the number additional terms as far as possible.*

2. I understand that the inland water level can't necessarily be simulated for the future climate, and the authors justify values/results might be different in the simulated climate control period because they're using a different metric – drainage system overload. A question I have is, can't the drainage system overload be calculated for the observational data in the same way and compare the types of events they see driving drainage system overload versus inland water level difference (what is currently calculated?) This could be helpful in distinguishing whether any variation in driving events is due to the different metric or is due to the model simulations.

   *Thanks for this suggestion. In principal, this is a good point, if data in sufficient resolution would be available. Unfortunately, the observed local climate data (in daily resolution) are not available in required temporal resolution (hourly). Thus, a disaggregation would be required introducing additional uncertainty. We added a comment on this in section 2.4. However, the system overloads calculated for the control period fit well to the recent experience of the regional water boards. We added an explanation on this in section 3.2.1. to emphasize the plausibility of the simulation results for the control period.*

3. Figure 4 shows that some of the highest sea levels, precipitation events, and inland water levels co-occur (where blue, red, and orange align). The MPI model shows this as well (using the drainage capacity metric) while the HadGEM model rarely shows this. Can this be physically explained? Does this mean the HadGEM model is actually not a great representation of the climate system?

   *We would be carefully to rate whether HadGEM is able or not to reproduce the climate system just based on this compound characteristic. However, it is remarkable that the 15 highest system overload events in the control period simulated by MPI occur with sea levels which are more than 80 cm higher than by HadGEM. Thus, for the HadGEM model, with similar cumulative rainfall amounts, the antecedent saturation of the soil (before intense rainfall events) seems to have a greater impact for HadGEM simulations. We added some discussion on this different behaviour in section 3.2.1. in order to emphasise model specific behaviour.*

4. Following on, figures 4, 5, 6, 11, and 12 require the reader to estimate how many "extreme" precip, sea level, and inland water level/drainage capacity co-occur. It would be interesting to actually quantify the number of drainage capacity/inland water level extremes that do NOT occur versus those that do occur with the driving variables, which would more easily pull out the statistics/comparisons of co-occurrence across the plots.

   *This is a good suggestion.*
   *In the revised manuscript we provide numbers for the different kind of events as requested to help the reader distinguishing between the different types (sections 3.2.1, 3.2.4). This might help to assess the importance of compound events.*

Discussion:

5. The manuscript results are very specific to the location. A broader discussion of the results and how they may relate or be extended to other locations should be added to ensure the overall findings relate to the broader readership of NHESS.

   *Thanks for this important suggestion. We are aware of several other studies with similar behaviour, such as from the Netherlands, Great Britain and from China. We added a a few sentences on the possible extension to the locations to the discussion section discussing regional transferability.*

6. I was expecting more of a discussion on the variation across the future climate model results. The discussion presents the future scenario results as very similar to each other, but I see distinct differences. For example, why does sea level matter more for the MPI events than it does for the HadGEM?

   *In the original manuscript, we were hesitating to discuss the projections in too much detail due to the small ensemble of combinations of climate models and scenarios. We focused on the general characteristics of the generation of compound events.*
   *But indeed, the scenarios themselves differ, and sea level has a different impact on compound events calculated by MPI and HadGEM models.*
   *One reason is that MPI simulates highest precipitation rates during the storm season in late autumn, while HadGEM also simulates extreme and precipitation events in winter (mentioned already in section 3.2.4.), associated with different event characteristic (e.g., in rainfall duration). We evaluated those events in more detail for the revised manuscript.*

7. The timing (during the year) of precipitation events is highlighted in the results and discussion, but there's no similar analysis for sea level. Do high sea level events tend to coincide in the same months as high precipitation? If they do not, is this the reason there aren't more extreme sea level and precipitation events combining?

   *For the climate scenario RCP8.5 we found a slight increase for storm events in fall (for the MPI model) and in winter (for the HadGEM model). A new figure has been added to illustrate this trend.*
   *The largest compound events for MPI are associated with sea levels which are about 0.5 m higher than for the HadGEM model. This seems to be an important aspect which was added to the discussion.*
   *In addition, we analysed the timing of the 15 hightest events for the different models and scenarios and provide insight in the results in section 3.2.4*

**Specific comments**

1. Sea levels need a datum specified in all plots

   *We added the datum to all figure captions (m above NHN; "standard elevation zero")*

2. In the introduction different types of compound events are defined, but only one is considered here. I'm not sure all need to be brought up.

   *This is a good point, mentioned by the other reviewers, as well.*
   *We used the classification only implicitly for the discussion of the explanatory power of our model approach. In a first attempt, we assume that the multivariate mechanism can explain the pattern of the events. But this seems to be not sufficient, at least preconditions such as antecedent soil moisture in the marsh also affect the pattern. We therefore now apply the classification explicitly in the discussion of the identified compound events in the results and discussion sections.*

3. Line 32: "Zealand" spelled incorrectly

   *We corrected this typo.*

4. Line 43: It is not clear what is meant by "individual events mentioned above" is. One at a time? The connection isn't clear.

   *We adjusted the sentence to make it more explicit and avoid misunderstanding.*

5. Line 57: I believe the Svensson and Jones 2002 article didn't identify more compound flood events, but a higher dependence between compounding forcing variables. They didn't model flooding.

   *Thanks for this correction. We adjusted the text accordingly.*

6. Line 88: How extreme are we talking?

   *This is a good point. We explain this in detail in section 2.4.3 and in figure 3. Pumps need to be switched off from 3.75 m geodetic head upwards.*

7. Line 115: How well does the bias correction do – especially with the extreme data as that is the focus of this study?

   *Quality of bias correction for the region is described in detail by Bormann and Kebschull (2023) as well as by Ley et al. (2023). We added the citations to the text.*

8. The precipitation metric is 3-day precipitation and the sea level metric is an average over 5 tidal periods. While this was used in previous literature that is cited for a specific storm event and region, this manuscript needs a justification as to why those metrics are used and relevant for this location.

   *Thanks for this advice. The 3 day period is chosen based on literature (van den Hurk et al., 2015) and the experience of the regional water board members. In the past, coastal drainage infrastructure was dimensioned based on two day extremes. Since already three day storm series are observed, they aim at adaptation planning for extreme events with a duration of three days. We add this justification to section 2.3.*

9. Line 156: the inland water level is assumed to be constant for the drainage capacity calculations. How good of an assumption is this? How different are the results if the inland water level is varied?

   *As shown in figure 4, the maximum deviation from this assumption in the available 20 years of observation was 44 cm. Such deviations only occur in case of system overload. Usually, deviations are significantly smaller and do not have a strong influence on pumping rates. We added a short justification to section 3.1.*

10. - Values of future sea level projections at mid or end of century  would be helpful!

    *We provided average projections in Figure caption of Figure 8.*

11. Line 226: It's mentioned that sea level and precipitation don't represent the sole drivers of inland water anomalies – what is meant by this? Nonlinearities? Other variables? This could be expanded upon in the discussion section.

    *We explain this while relating our case to the typology of compound events. One important driver is the antecedent soil moisture (saturation of the area prior to the event). Unfortunately, soil moisture is not measured in the region, thus a quantitative evaluation is not feasible.*

12. Line 380: will "not be"

    *We corrected this typo.*

13. How many realizations of each different climate model was used? For example is the HadGEM results representative of one end member or an ensemble? And if an ensemble, how many endmembers were used?

    *There was just one realisation per climate model available and used in hourly resolution. Information was added to section 2.2.*

14. Figure 1: Needs an improved legend or caption. There are lots of components of the map that aren't well defined. Needs datum for elevation.

    *Explanation in the figure caption was revised. The datum has been added (m above NHN; "standard elevation zero").*

15. Figures 4,5,6,11,12: Can the numbers within the circles be explained somewhere in the captions? Sea levels need datum.

    *The numbers in the circles are now explained explicitly in the figure captions. Datum has been added (m above NHN; "standard elevation zero").*

16. Figure 10: I find it pretty interesting that for the MPI the days with system overload are really similar for mid century 8.5 and end of century 2.6!

    *We also recognised this, but we don't have a physical explanation for this similarity so far.*

---

## Author Response (AR2)

Dear editor, dear referees,

Thanks once more for your additional helpful comments and corrections. We adjusted the final manuscript as follows.

**Anonymous referee #2**

237 "resulting" or "which resulted"

*We corrected this typo (resulting).*

314 I am glad to see "storm tide" defined now but it comes a bit late and is used several times before.

*You are right. We moved the definition to the introduction (new line 47) where we use the expression for the first time.*

**Anonymous referee #3**

It looks like there's a definition of extreme, intense/high, and moderate events in the tracked changes version of the manuscript, but not in the clean submitted manuscript. This made it hard to interpret Table 1 (and left me wondering why line 175 was its own paragraph.) Please fix.

*Thanks a lot for this important comment. This happened probably while accepting all changes in Word. We manually added the missing sentence.*

I think the authors need to be clearer during the definition of extreme, moderate, and intense/high, that what falls into one of these categories magnitude-wise is not the same in the future. They address this near line 415, but I think this could be mentioned up front when defining the terminology. For example, line 491 "…as long as rainfall amounts and intensities will not exceed those represented by the climate projections, moderate rainfall alone – without any restriction in drainage capacity – is not expected to lead to large flooding… until the end of 21st century" is unclear whether the term "moderate" relates to the events in observations, near future, far future, etc., when moderate would range from <20 mm in observations and <40/50mm in the end of century model results (from what I could interpret the definition to be). I do think some definition based on magnitude would be easier to interpret than how the authors have chosen to define here but will leave it to them to decide.

*Thanks for this critical comment. This is a general problem of classification approaches to non-stationary systems. A 100-runoff year event from the past will probably not be a 100-year event in the future if the climate changes. We define and apply our approach as a relative one, based on sorting the events according to their magnitude. Thus, an extreme event from the past could be only an intense event in the future. That's correct. However, the advantage of our approach is that we can assess the contribution of events to the*

*magnitude of a compound event relative to the ensemble of the respective period. This would be affected by using a classification based on absolute values. Therefore we decided to keep our approach. To take you critique info account, we add a sentence on this at the end of chapter 2.3*

As a reader from outside Germany, I think it would be helpful to have the definition of a water board introduced earlier – maybe when first mentioned in the introduction, rather than in Line 90.

*We moved the definition of the water boards to the introduction as suggested.*

Line 71: "Other analyses" a little vague. Other analyses of compound flooding?

*Changed to "other climate change impact analyses"*

Line 160: Any deviation? Or is it a positive deviation for water system overload.

*Thanks for this specification. Yes, we talk about a positive deviation. We adjusted this expression.*